# Structural diversity of axonemes across mammalian motile cilia

Miguel Ricardo Leung[1,7,8], Chen Sun[2,8], Jianwei Zeng[2,8], Jacob R. Anderson[3,8], Qingwei Niu[2,4], Wei Huang[5], Willem E. M. Noteborn[6], Alan Brown[3✉], Tzviya Zeev-Ben-Mordehai[1✉] & Rui Zhang[2✉]

Reproduction, development and homeostasis depend on motile cilia, whose rhythmic beating is powered by a microtubule-based molecular machine called the axoneme. Although an atomic model of the axoneme is available for the alga *Chlamydomonas reinhardtii*[1], structures of mammalian axonemes are incomplete[1–5]. Furthermore, we do not fully understand how molecular structures of axonemes vary across motile-ciliated cell types in the body. Here we use cryoelectron microscopy, cryoelectron tomography and proteomics to resolve the 96-nm modular repeat of axonemal doublet microtubules (DMTs) from both sperm flagella and epithelial cilia of the oviduct, brain ventricles and respiratory tract. We find that sperm DMTs are the most specialized, with epithelial cilia having only minor differences across tissues. We build a model of the mammalian sperm DMT, defining the positions and interactions of 181 proteins including 34 newly identified proteins. We elucidate the composition of radial spoke 3 and uncover binding sites of kinases associated with regeneration of ATP and regulation of ciliary motility. We discover a sperm-specific, axoneme-tethered T-complex protein ring complex (TRiC) chaperone that may contribute to construction or maintenance of the long flagella of mammalian sperm. We resolve axonemal dyneins in their prestroke states, illuminating conformational changes that occur during ciliary movement. Our results illustrate how elements of chemical and mechanical regulation are embedded within the axoneme, providing valuable resources for understanding the aetiology of ciliopathy and infertility, and exemplifying the discovery power of modern structural biology.

Motile cilia are used by unicellular and multicellular organisms either to propel themselves through fluid or to move fluid across their surfaces. Ciliary motility is driven by a microtubule-based supramolecular assembly known as the axoneme, which consists of nine doublet microtubules (DMTs) surrounding a central apparatus of two singlet microtubules. DMTs are patterned into repeating 96-nm units by two rows of dynein arms (outer dynein arms (ODAs) and inner dynein arms (IDAs)), up to three T-shaped mechanoregulatory complexes called radial spokes (RSs), the nexin–dynein regulatory complex (N-DRC) that links neighbouring DMTs and a network of coiled coils that regulates the docking and periodicity of the aforementioned complexes. In addition, the DMT lumen is extensively decorated with microtubule inner proteins (MIPs) that bind in varying multiples of the 8-nm tubulin repeat, but with an overall periodicity of 48 nm that is in coherent register with the external 96-nm repeat.

Over the past 20 years, cryoelectron tomography (cryo-ET) and cryoelectron microscopy (cryo-EM) have brought our understanding of the axoneme to the molecular level, culminating in a recent atomic model of the 96-nm modular repeat from the green alga *Chlamydomonas reinhardtii*[1,6]. However, corresponding models of mammalian axonemes are incomplete[1–5]. For instance, the model of a human DMT from respiratory cilia[1] lacks RS3, a prominent complex present in most ciliated organisms but absent from *Chlamydomonas*, and does not account for many enzymes or regulatory kinases thought to be anchored to the axoneme[7].

Cryo-EM and cryo-ET have also shown marked variation in axonemal subcomplexes across species and cell types[1–3,8–12]. This variation reflects the diversity of ciliary form and function in nature, and even within an organism; for instance, ependymal cilia in brain ventricles drive the flow of watery cerebrospinal fluid, whereas respiratory cilia in the trachea propel viscous mucus along the airway surface. Epithelial cilia and sperm flagella have distinct waveforms[13] and vary greatly in length, ranging from a few microns in the respiratory tract to tens or even hundreds of microns in sperm. They also respond differently to mutations in proteins that they are proposed to share. However, the

[1]Structural Biochemistry Group, Bijvoet Centre for Biomolecular Research, Utrecht University, Utrecht, the Netherlands. [2]Department of Biochemistry and Molecular Biophysics, Washington University in St. Louis, School of Medicine, St. Louis, MO, USA. [3]Department of Biological Chemistry and Molecular Pharmacology, Blavatnik Institute, Harvard Medical School, Boston, MA, USA. [4]Department of Cell Biology & Physiology, Washington University in St. Louis, School of Medicine, St. Louis, MO, USA. [5]Department of Pharmacology, Case Western Reserve University, Cleveland, OH, USA. [6]Netherlands Centre for Electron Nanoscopy (NeCEN), Leiden University, Leiden, the Netherlands. [7]Present address: Hubrecht Institute-KNAW & University Medical Center Utrecht, Utrecht, the Netherlands. [8]These authors contributed equally: Miguel Ricardo Leung, Chen Sun, Jianwei Zeng, Jacob R. Anderson. ✉e-mail: alan_brown@hms.harvard.edu; z.zeev@uu.nl; zhangrui@wustl.edu

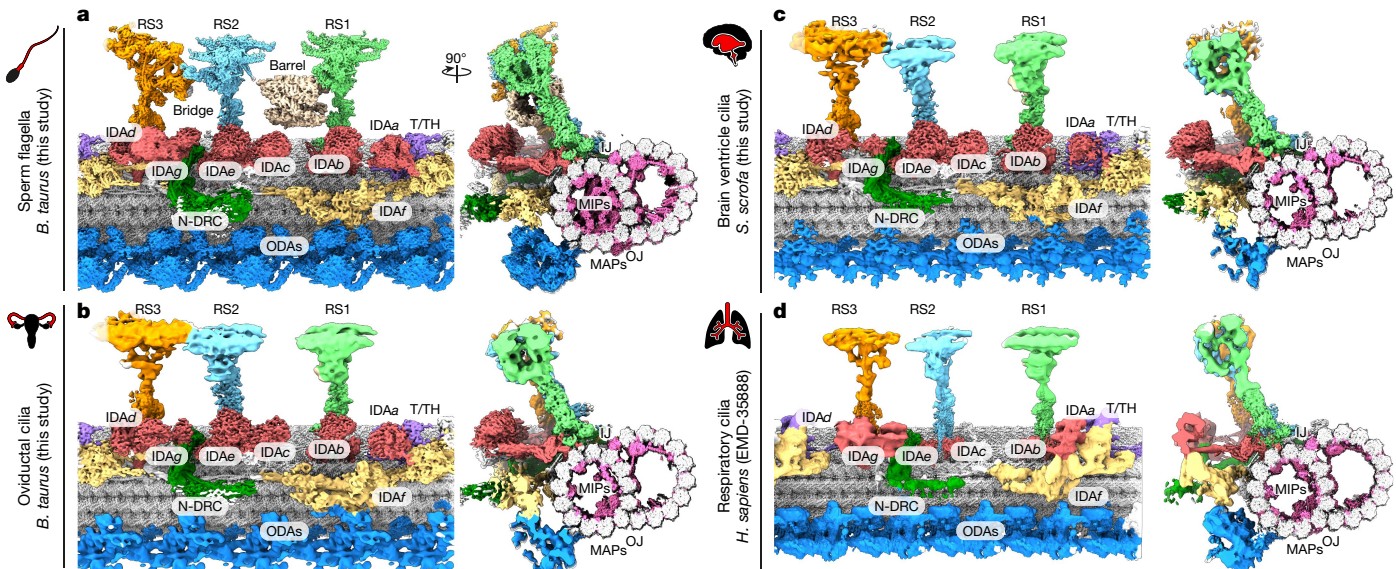

**Fig. 1 | Cryo-EM reconstructions of the 96-nm axonemal repeat of motile cilia from different mammalian cell types.** Each panel shows a longitudinal and cross-sectional view of a composite cryo-EM map of a 96-nm repeat unit of a doublet microtubule from bovine sperm flagella (**a**), bovine oviductal cilia (**b**), porcine brain ventricle cilia (**c**) and human respiratory cilia (**d**). The reconstruction in **d** is EMD-35888 (ref. 1). Each major axonemal complex is given a unique colour with the doublet microtubule in grey. IJ, inner junction; MAP, microtubule-associated protein; OJ, outer junction.

lack of high-resolution structures of axonemes from different mammalian cell types prevents a full understanding of how differences in individual proteins or protein complexes contribute to ciliary diversity in normal function and in disease.

## Comparison of epithelial and sperm DMTs

To shed light on the structural diversity of axonemes across different mammalian motile-ciliated cell types, we used single-particle analysis (SPA) cryo-EM to reconstruct the native 96-nm repeat of DMTs from disintegrated axonemes of sperm flagella (*Bos taurus*) and epithelial cilia isolated from either the oviduct (*B. taurus* and *Homo sapiens*) or brain ventricles (*Sus scrofa*) (Fig. 1, Extended Data Fig. 1a–d, Supplementary Figs. 1–7, Supplementary Tables 1, 2 and Methods). Separately, we reconstructed the 96-nm repeat from intact porcine (*S. scrofa*) oviduct cilia using cryo-ET and subtomogram averaging, showing consistency with our SPA structures, especially near the microtubule surfaces (Extended Data Fig. 1e–g and Methods). By comparison of these reconstructions with published maps of human respiratory cilia[1], we define how the structure of the axoneme varies across motile-ciliated cell types of the mammalian body.

Our work demonstrates that the DMTs of multiciliated epithelial cells are almost structurally indistinguishable, with differences restricted to the intraluminal tektin bundle and associated proteins RIBC1/2 (Extended Data Fig. 2). The overall similarity of epithelial DMTs reflects the similarity of epithelial cilia in general—they are all approximately 5–10 µm long, consist of an axoneme sheathed by a ciliary membrane and have similar waveform dynamics. Nevertheless, the absence of obvious structural specializations in DMTs from epithelial cilia is somewhat unexpected considering their roles in propelling liquids of very different viscosity, and the different sensitivities of tissues to ciliopathic mutations. For example, genetic ablation of the β-tubulin isotype TUBB4B causes severe loss of tracheal and oviductal cilia in mice, but has no apparent effect on the number, length or beat frequency of brain ependymal cilia[14]. Our structural and proteomic data confirm that TUBB4B is the main β-tubulin isotype of pig ependymal DMTs—as it is in all motile cilia examined (Supplementary Tables 3 and 4)—suggesting that differential sensitivity to

TUBB4B depletion cannot be explained solely by gross differences in DMT structure.

In contrast to the relatively homogeneous structures of epithelial DMTs, direct comparison of bovine DMTs from three different tissues shows that sperm DMTs have an additional layer of complexity (Fig. 1) that extends to the MIPs that decorate the lumen of axonemal DMTs[2,3] (Extended Data Fig. 2). Our structures further show that ciliary microtubule-associated proteins (CIMAPs) bound close to the external surface of the DMT[2,15] are ubiquitous features of mammalian axonemes but have cilium-specific distribution (Extended Data Fig. 3a,b). For example, CIMAP3 is present in all mammalian axonemes hitherto studied, yet CIMAP2, which binds the same protofilament cleft, is found only in sperm (Extended Data Fig. 3b). These structural observations are supported by both proteomics (Supplementary Table 4) and expression data[16].

## Model of the sperm 96-nm modular repeat

To define the molecular nature of sperm-specific axonemal specializations, we used our cryo-EM and proteomics data along with artificial intelligence-enabled modelling to build an atomic model of the 96-nm repeat of the bovine sperm DMT (Supplementary Table 5). This represents a highly complete atomic model of a mammalian axonemal DMT. The rationale for assigning individual proteins, including identification strategies and supporting evidence from the literature, is summarized in Supplementary Figs. 8–33. Note that this model represents a consensus of all nine DMTs, because information about their spatial organization is lost during sample preparation for SPA. We identify 34 additional axonemal proteins compared with recent models of human respiratory cilia[1] and the sperm 48-nm repeat[2]. Based on structural, proteomic and expression[16] data, we assign 21 of these proteins as conserved across cell types and 13 as sperm specific (Fig. 2a and Supplementary Table 6). We identify new proteins across nearly every axonemal complex, including an ARMH1 subcomplex that is distributed asymmetrically around the axoneme based on in situ cryo-ET data[10] (Extended Data Fig. 4a,b); WDR64 that binds atop the CCDC96/CCDC113 heterodimer (Extended Data Fig. 4c); N-DRC proteins LRRC74A and ANKEF1, the latter being positioned to interact

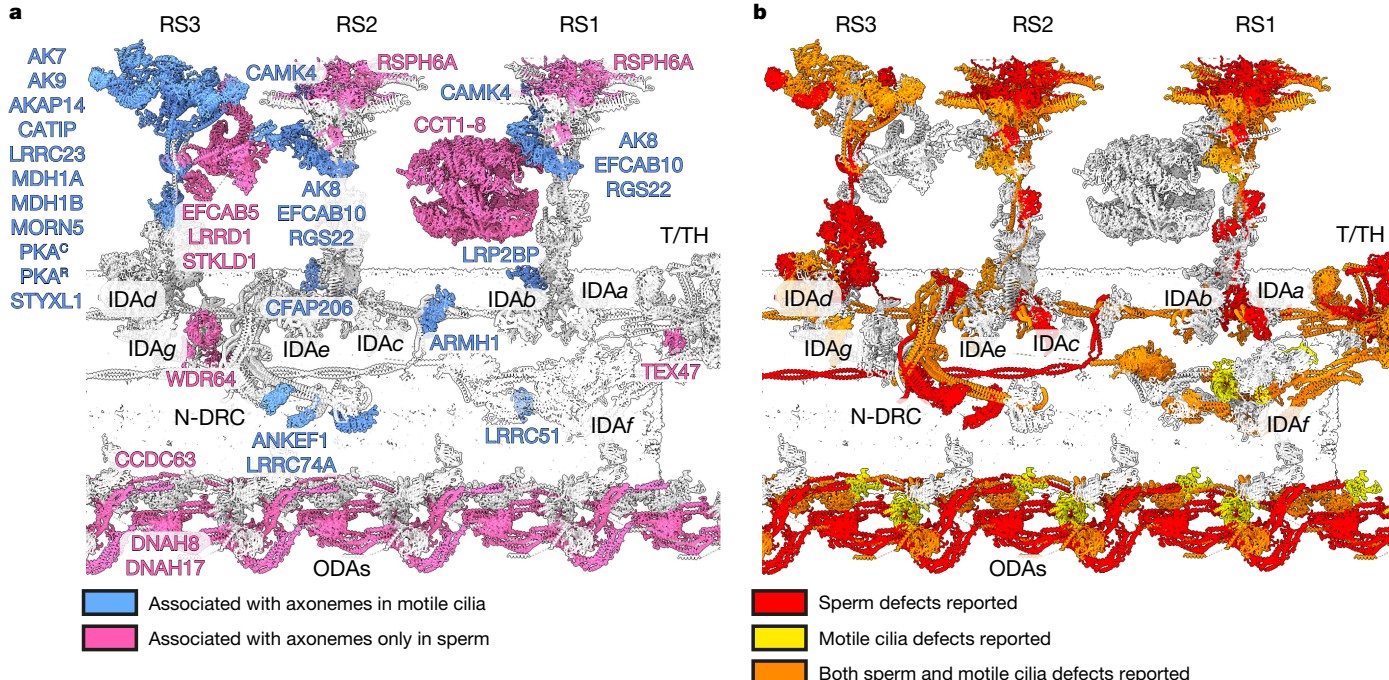

**Fig. 2 | Newly identified sperm-specific and disease-linked axonemal proteins. a**, Thirty-four newly assigned axonemal proteins are identified in this study, including 21 'general' proteins that are conserved across cell types (blue) and 13 that are sperm specific (pink). Note that CCDC63, DNAH8, DNAH17 and RSPH6A were previously found to be sperm-specific axonemal proteins but are included here for emphasis. **b**, Proteins implicated in infertility and other motile ciliopathies are colour coded by whether their disruption in humans or model organisms causes defects in sperm flagella (red), other motile cilia (yellow) or both (orange).

with the neighbouring DMT (Extended Data Fig. 4d); and LRRC51 and CFAP206, which contribute to the docking of inner dynein arm *f* (IDA*f*) and RS2, respectively (Extended Data Fig. 4e,f). Mapping the locations of proteins implicated in motile ciliopathy or infertility onto our model (Fig. 2b and Supplementary Table 7) demonstrates that genetic lesions in almost any subcomplex can lead to disease, emphasizing the intricate interconnectivity of the axoneme.

## Molecular composition of RS3

We build an atomic model for RS3 (Fig. 3 and Supplementary Video 1) that accounts for most of the densities resolved in our maps, and also by in situ cryo-ET[9,10] (Extended Data Fig. 5a). Many of the 17 proteins we assign to RS3 have enzymatic domains, including kinases, phosphatases and dehydrogenases (Extended Data Fig. 5b,c). These proteins are distinct from those in RS1 and RS2, which are otherwise similar to each other and consist mainly of non-enzymatic structural proteins. Proteins that constitute RS3 (with the exception of a sperm-specific RS2–RS3 bridge) are conserved across mammalian motile cilia (Supplementary Table 6) and have orthologues in ciliated organisms that have RS3. Underscoring their importance for ciliary structure and function, mutations in several RS3 proteins lead to ciliopathy or infertility, including STYXL1, CATIP and LRRC23 (Fig. 2b and Supplementary Table 7).

The C-terminal helical domain of CFAP91, one of the base proteins of RS3 (ref. 1), plays a central role in RS3 assembly by serving as a platform onto which many other proteins dock (Supplementary Video 1). This typically occurs through RIIa or DPY30 domains, similar to how RS1/RS2 proteins assemble around RSPH3. However, unlike RSPH3, which is present in two copies leading to the RS1/RS2 heads having pseudo-twofold symmetry, CFAP91 is present in only one copy and thus the RS3 head is asymmetric (Fig. 3).

Our structure also shows that the sperm-specific bridge between RS2 and RS3 (ref. 9) is formed by an interaction between RGS22 in the head of RS2 and EFCAB5, a sperm-specific protein in the stalk of RS3 (Fig. 3

and Extended Data Fig. 6). Binding to EFCAB5 appears to induce a conformational change in RGS22, which adopts a more extended conformation in RS2 than in RS1 (Extended Data Fig. 6b). Sperm-specific RS3 proteins LRRD1 and STKLD1 interact with EFCAB5 to further stabilize the structure. A globular density binds to STKLD1 to complete the bridge, but flexibility prevented us from obtaining reconstructions sufficient to identify it (Extended Data Fig. 6a). Interspoke linkages, such as the RS2–RS3 bridge, may functionally couple RSs to one another, constraining their tilting during ciliary beating or facilitating mechanical propagation of signals across the spoke network. Variation in RS–RS interactions within and across species—such as in *C. reinhardtii* in which RS1 and RS2 interact directly[6]—may therefore contribute to cell-specific fine-tuning of the ciliary beat.

## An RS-associated ATP regeneration system

Each beat cycle of an axoneme consumes roughly 230,000 molecules of ATP[17]. Simulations suggest that this enormous demand cannot be met by simple diffusion alone, especially for long cilia in which the most distal dyneins are far from mitochondria, the main source of ATP[18]. To maintain rhythmic beating, cilia contain mechanisms for regeneration of ATP and depletion of ADP. One such mechanism uses adenylate kinases (AKs), which can regenerate ATP and AMP from two molecules of ADP by catalysis of a reversible nucleotide phosphoryl exchange reaction[19,20]. AKs are essential for proper ciliary function, because disruption of cilia-enriched AKs can lead to human ciliopathy or infertility (Supplementary Table 7).

Despite the ubiquitous presence of AKs in motile cilia and their importance for ciliary motility, previous structures did not pinpoint their locations. Now, our structures show that three different AKs are anchored to the RSs of each mammalian DMT examined: AK8 is bound to the neck of RS1 and RS2, with AK9 and two copies of AK7 being found in the head of RS3 (Fig. 4). RS tethering is achieved through helical domains that dimerize with a neighbouring protein and dock onto

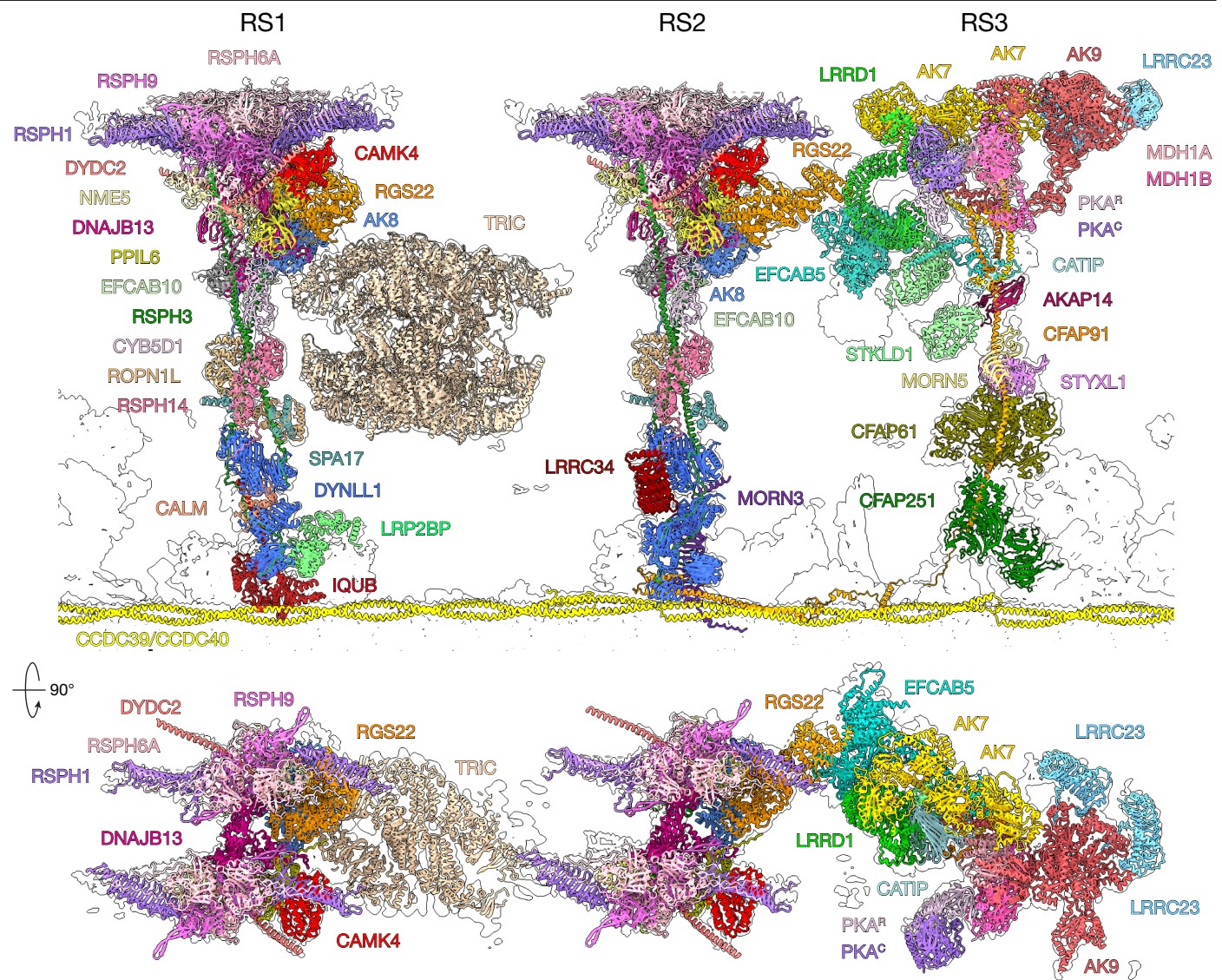

**Fig. 3 | Structures of RSs in mammalian sperm.** Atomic models of RS proteins (coloured) fitted into the cryo-EM density map from bovine sperm (outline).

an amphipathic helix formed by a central RS protein (Extended Data Fig. 7a–c). In the case of AK8, its RIIa domain dimerizes with EFCAB10 and docks onto RSPH3 (Extended Data Fig. 7b). The similar DPY30 domain in AK7 dimerizes with either EFCAB5 (in AK7-A) or AK9 (in AK7-B); both dimers then dock onto CFAP91 (Extended Data Fig. 7c). Tethering of AKs to RSs ensures that they are in close physical proximity to ATP-consuming, ADP-generating dynein motors and are uniformly distributed along the length of the axoneme to maintain consistent ATP levels throughout the cilium.

The three RS-associated AK isozymes have a variable number of catalytic domains (one in AK7, two in AK8 and four in AK9), providing a total of ten catalytic domains every 96 nm (Extended Data Fig. 8). AKs undergo major conformational changes during their catalytic cycle, assuming 'closed' and 'open' states associated with catalysis and product release, respectively[21]. Comparison of our models with the crystal structures of AK in different states[22,23] shows that the catalytic domains of axonemal AKs adopt a mix of open and closed states (Extended Data Fig. 8c). For instance, whereas one catalytic domain of AK8 is in the closed state, its second appears to be stabilized in an open state by its neighbouring proteins DNAJB13, PPIL6 and RGS22 (Extended Data Fig. 7b). Interestingly, we observe density consistent with small molecules in the nucleotide-binding pockets of both open

and closed catalytic domains (Extended Data Fig. 8c). The open conformations of catalytic domains I and IV of AK9 may accommodate the binding of LRRC23 (Extended Data Fig. 8c), a protein required for assembly of the RS3 head and therefore necessary for male fertility in mice[24]. Whether domain II of AK8 and domains I and IV of AK9 can cycle between open and closed conformations is unclear, but the remaining catalytic domains would, in principle, be able to transition between open and closed conformations during catalysis. Relating the catalytic activity of the AKs to changes in their conformations and interactions within the context of the axoneme will be an interesting, if challenging, avenue for future work.

## RS-anchored signalling protein kinases

We find a protein kinase A (PKA) holoenzyme tethered to the head of RS3 (Fig. 3 and Extended Data Fig. 7a,e). PKA is a cAMP-dependent kinase implicated in the regulation of ciliary beat frequency in response to increased cAMP levels in both epithelia[25] and sperm[26]. In mammalian sperm, PKA signalling is central to the hyperphosphorylation cascade and motility activation characteristic of capacitation in the female reproductive tract[27]. Anchoring PKA to RS3 provides a spatial mechanism by which it could rapidly alter the phosphorylation state

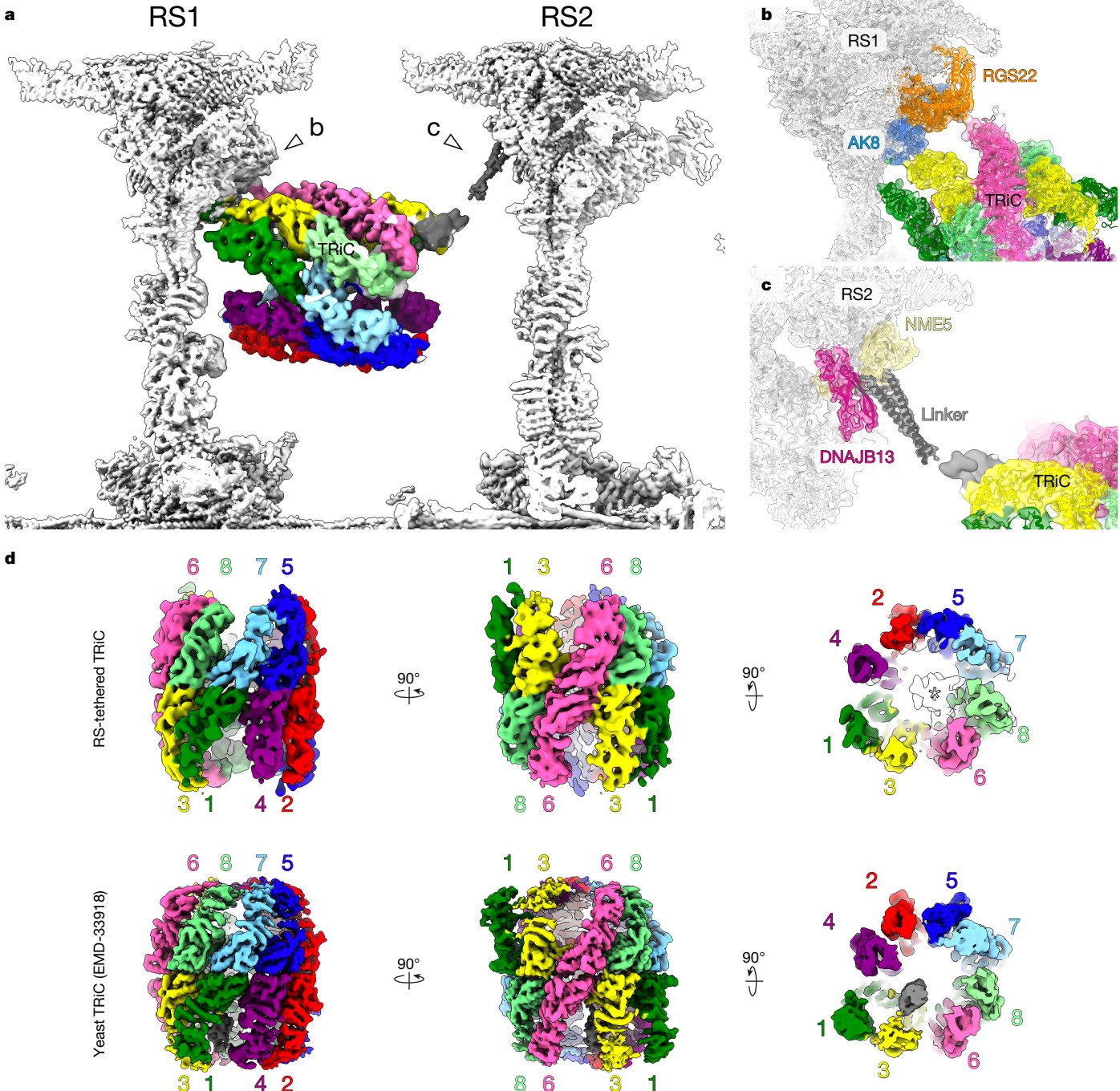

**Fig. 4 | The sperm-specific barrel is an RS-tethered TRiC chaperone. a**, TRiC is suspended between RS1 and RS2. **b**, TRiC contacts RS1 by small interfaces with RGS22 and AK8. **c**, TRiC is tethered to RS2 by a hitherto unidentified linker protein (grey) anchored to DNAJB13 and NME5. **d**, Assignment of subunit order in the barrel, based on the inherent asymmetries of TRiC, demonstrated by comparison with the structure of yeast TRiC[33]. Asterisk marks an unidentified luminal density of the distal ring (closer to RS2) in RS-tethered TRiC. This density binds at a location different from the binding site of the cochaperone Plp2 in yeast TRiC (grey density)[33].

of axonemal proteins, thereby leading to changes in ciliary beat frequency. Indeed, phosphorylation of an axonemal dynein light chain is associated with activation of sperm motility in fish and sea urchins[28].

Unexpectedly, the RS3-bound PKA enzyme is a dimer consisting of one copy of a regulatory subunit (PKA$^R$) bound to one copy of a catalytic subunit (PKA$^C$), although we cannot determine specific isoforms at the (approximate) 5 Å resolution of our maps. This stoichiometry differs from the tetrameric arrangement observed in crystal structures[29]. PKA$^R$ interacts directly with a malate dehydrogenase dimer in the middle of the RS3 head, but PKA$^C$ interacts directly only with PKA$^R$

(Extended Data Fig. 7e). By tethering PKA to RS3 entirely through its regulatory domain, the catalytic subunit is free to dissociate from PKA$^R$ and diffuse within cilia in response to cAMP. Super-resolution microscopy of mouse sperm shows that PKA$^C$ distribution changes from a tight cylinder of (approximate) mean radius 70 nm in non-capacitated sperm to a broader (approximate) 105-nm radius in capacitated sperm[30].

We identify two additional RS-tethered protein kinases (CAMK4 and STKLD1) (Fig. 3). CAMK4, a member of the calcium/calmodulin-dependent protein kinase family, is bound to both RS1 and RS2, in which it nestles into a pocket surrounded by RSPH1, DYDC2 and

PPIL6 (Extended Data Fig. 7d). Density resembling an ADP molecule can be resolved in the nucleotide-binding pocket of both copies of RS-associated CAMK4, suggesting that the kinase is active (Supplementary Fig. 14); indeed, CAMK antagonists inhibit human sperm motility[31] and genetic variants have been linked to human infertility (Supplementary Table 7). STKLD1, an uncharacterized sperm-specific kinase, is bound to the stalk of RS3, where it interacts with EFCAB5 and a hitherto unidentified density to form part of the sperm-specific RS2–RS3 bridge (Extended Data Fig. 6a).

## Sperm-specific RS-bound TRiC chaperone

The most prominent difference between mammalian sperm flagella and epithelial cilia is the presence of a sperm-specific, barrel-shaped density suspended between RS1 and RS2 (Fig. 1). These structures were first observed by in situ cryo-ET[9,32], and subsequently shown to be distributed asymmetrically around the axoneme[10].

We resolve the barrel at roughly 7–8 Å, allowing us to unambiguously identify it as a TRiC chaperone (Fig. 4) consisting of two stacked rings of eight CCT subunits each. The intrinsic asymmetry of TRiC[33] allowed us to assign subunit order and to model how the chaperone complex interacts with RS1 and RS2 (Fig. 4a). The proximal ring (closer to the minus end of the DMT) binds to the head/neck of RS1 through small interfaces with RGS22 and AK8 (Fig. 4b). The distal ring (closer to the plus end) is tethered to RS2 by a linker anchored to DNAJB13 and NME5 in the RS2 head (Fig. 4c). Luminal density within the distal ring cannot be assigned due to limited resolution, but it is bound in a different location from the yeast TRiC cochaperone Plp2 (ref. 33) (Fig. 4d).

Identification of an RS-tethered TRiC chaperone is consistent with proteomics data indicating that all CCT subunits are highly abundant in mammalian sperm, and with immunofluorescence data localizing CCT subunits along the mammalian sperm flagellum[34–36]. Congruent with the presence of RS-tethered TRiC only in mammalian sperm, CCT subunits are not robustly detected in epithelial motile cilia; for instance, we detect only CCT8 in human and bovine respiratory axonemes (Supplementary Tables 4 and 6). It is also possible that the protein/s tethering TRiC to RS2 (Fig. 4c) are expressed only in sperm, further explaining their prominent, regular arrangement in sperm but not in epithelial axonemes.

## Dynein prestroke state conformations

Ciliary motility depends on each of the thousands of dynein motors distributed along the length of the axoneme generating force through a powerstroke mechanism. Previous SPA studies have captured axonemal dyneins in a poststroke state[37–39], whereas cryo-ET studies have provided lower-resolution (around 30 Å) structures of the prestroke state[40–42]. However, a precise molecular understanding of how axonemal dyneins and their interactions change during a mechanochemical cycle remains elusive. Here we took advantage of our large dataset of particles and used three-dimensional classification to identify conformations of ODA and IDA*f* that resemble the prestroke state observed in situ[40] (Fig. 5, Extended Data Figs. 9a, 10 and Supplementary Videos 2, 3).

This prestroke state of IDA*f* can be particularly well resolved using our bovine sperm data (roughly 5 Å), allowing us to build an atomic model (Extended Data Fig. 9b,c) in which the conformations of the linker and stalk domains are consistent with a prestroke state[43] (Extended Data Fig. 9d). Compared with human IDA*f* in the poststroke state[1], the motor domains have shifted by around 8 nm towards the minus end of the axoneme and approach the DMT surface (Fig. 5 and Supplementary Video 2). The molecular contacts between the AAA+ rings of the *fα* and *fβ* motor domains are also remodelled, from an almost parallel configuration in the poststroke state to an almost perpendicular configuration in the prestroke state (Extended Data Fig. 9e). The movement of *fβ* pulls with it the IDA*f*-associated tether/tetherhead (T/TH) complex,

lifting it away from the DMT surface (Extended Data Fig. 9f and Supplementary Video 2). In this conformation, the β-propeller domains of the T/TH subunit, CFAP44, engage the motor domain of IDA*a* (DNAH12) (Extended Data Fig. 9f–h), which has also shifted towards to the minus end relative to its poststroke state (Fig. 5). TEX47, a sperm-specific subunit of the T/TH complex and orthologue of *Chlamydomonas* MOT7, directly interacts with the linker of *fβ* (Extended Data Fig. 9h).

In the prestroke state, *fα* and *fβ* interact with more proximal axonemal complexes (Fig. 5). First, the microtubule-binding domain of *fα* interacts with the intertwined helices of DRC9 and DRC10 of the N-DRC, providing a structural mechanism for coordination of two major regulatory complexes (Extended Data Fig. 9g, arrows). Second, the motor and stalk of *fβ* interact with the tails of the dynein heavy chains of IDA*d* and IDA*g*, respectively (Extended Data Fig. 9h). Given that the C-terminal helices of CFAP44 and CFAP43 also form part of the docking complex of IDA*d* and IDA*g*, our findings exemplify the importance of IDA*f* and its associated T/TH complex in regulating the coordinated activity of IDAs. Because IDA*f* shows fairly low motor activity in vitro[44], its function as a mechanochemical regulator may indeed be its major role in the axoneme. Because the microtubule-binding domains of IDA*f* would not contact the adjacent DMT in the prestroke state, an important future direction will be to elucidate the molecular mechanics of its powerstroke in situ.

## Discussion

Our work demonstrates that, whereas DMTs of multiciliated epithelial cells are structurally similar, sperm DMTs have additional layers of complexity arising from the incorporation of around 30 proteins not found in epithelial cilia. Among the most notable sperm-specific additions are MIPs[2,3], kinases and a TRiC chaperone suspended between RS1 and RS2. This projected number is likely to be conservative, because some densities in our sperm DMT reconstruction remain unassigned due to low resolution caused by either flexibility or the asymmetric distribution of proteins around/along the axoneme. Specialization of the sperm DMT reflects the exceptionality of mammalian sperm flagella—they are at least one order of magnitude longer than epithelial cilia, have a different waveform and are stiffer due to the presence of accessory structures such as outer dense fibres and the fibrous sheath that surrounds the central axoneme.

Importantly, sperm-specific proteins are additional to the general axonemal proteins, suggesting that complexity in sperm DMTs arose through accretion of additional subunits during evolution rather than remodelling of the DMT proteome. There are only a few known occurrences of sperm-specific paralogues replacing proteins found in epithelial cilia: CIMIP2A and CIMIP2B in the lumen of the A-tubule[2,3]; RSPH6A replaces RSPH4A in the heads of RS1 and RS2 (ref. 45); dynein heavy chains DNAH8 and DNA17 replace DNAH5 and DNAH9 in ODAs[46,47]; and CCDC63 replaces ODAD1 in the ODA docking complex[48]. We propose that WDR64, which binds the CCDC96/113 coiled coil, is a sperm-specific paralogue of WDR49, based on their shared domain organization but differing expression profiles. The reason why epithelial and sperm DMTs have different paralogues for a few proteins remains unclear, even following analysis of the structures. For example, RSPH6A in sperm RSs and RSPH4A in epithelial RSs form strikingly similar interactions and do not engage any other tissue-specific protein.

The observation that sperm-specific proteins bind atop general axonemal proteins implies a hierarchical and subsequent incorporation of these proteins during cilium assembly. This hypothesis aligns with data from a recent RNA sequencing study[49], which showed that genes for general axonemal proteins are expressed earlier than those for sperm-specific axonemal components during spermatogenesis (Supplementary Table 8). The distinct expression profiles of general and sperm-specific axonemal proteins suggest a transcriptional program for sperm-specific axonemal proteins specifically induced during

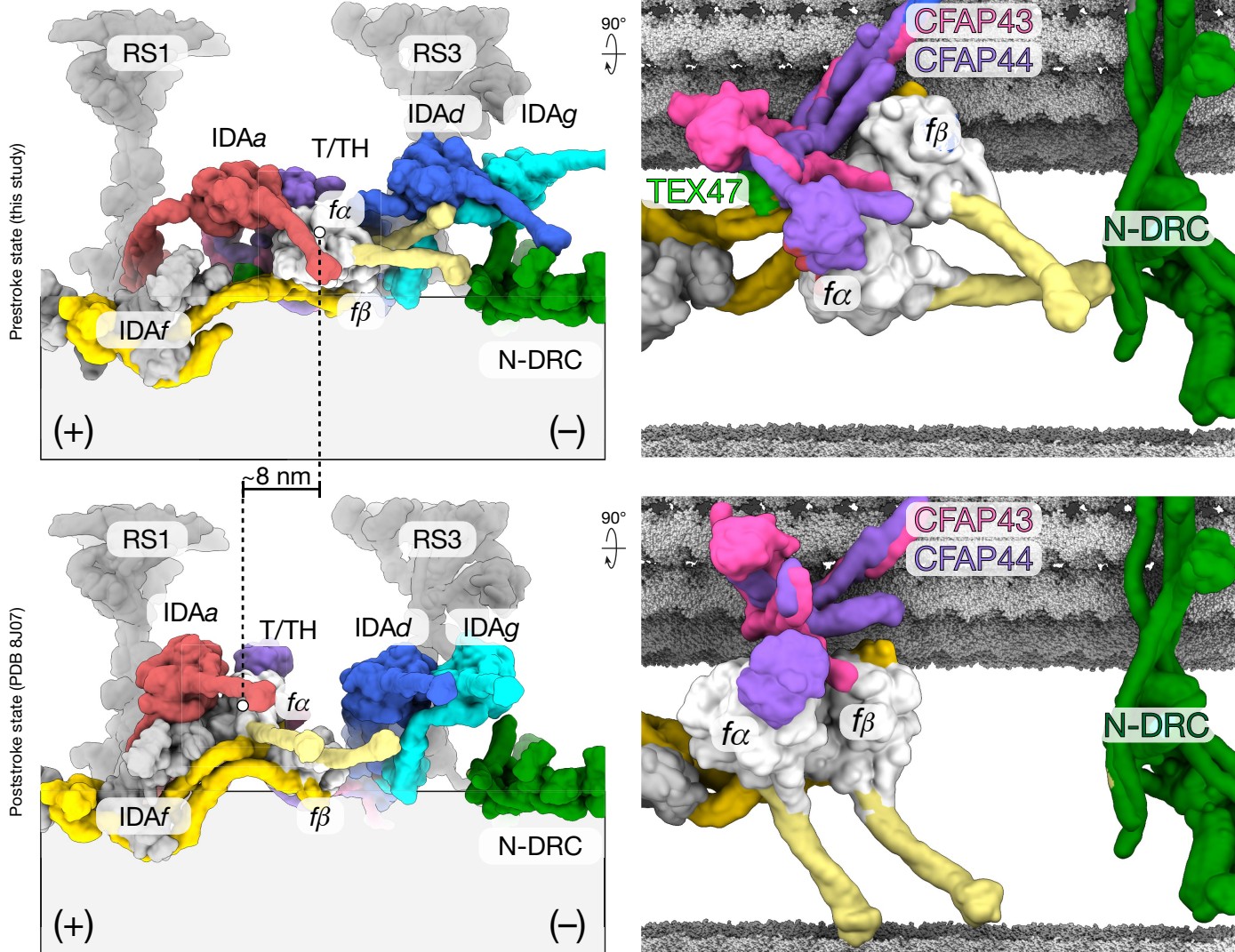

**Fig. 5 | Prestroke state conformation of IDA*f*.** Left, Molecular surface representations comparing the prestroke state resolved in this study (top) and the poststroke state resolved in EMD-35888 and modelled in PDB 8J07 (ref. 1) (bottom). Right, model showing how IDA*f* would interact with the B-tubule of the neighbouring DMT in pre- and poststroke states. A subtogram average from intact *Tetrahymena thermophila* axonemes (EMD-9023)[54] was used to model the position of the neighbouring DMT.

spermatogenesis. The mechanisms that trigger expression of these genes during spermatogenesis and suppress them in epithelial cell types require future study. Exogeneous expression of sperm-specific axonemal proteins in epithelial cilia offers a potential pathway to investigate how these proteins might contribute to generation of the unique waveform of sperm tails.

We show that TRiC is among the sperm-specific additions to the axoneme. As a chaperone for tubulin folding, TRiC contributes to the assembly of the extensive microtubule cytoskeleton of the ciliary axoneme and is required for ciliogenesis in *Tetrahymena*[50] and for spermiogenesis in *Planaria*[51]. Anchoring TRiC to sperm RSs could be important for construction of the exceptionally long microtubules of mammalian sperm, although both zebrafish and sea urchin sperm are of similar length yet neither have RS-associated TRiC[42,52]. TRiC could also play a mechanoregulatory role, restricting the motion of RS1/2 and coupling them to one another. In this case, the asymmetric distribution of TRiC around the sperm axoneme[10] could impart subtle differences in the bending properties of individual DMTs. Our work demonstrates the feasibility of identifying asymmetrically distributed proteins (such as TRiC and the ARMH1 complex) from large SPA datasets, guided by information from in situ cryo-ET.

Another intriguing possibility is that RS-tethered TRiC may function to locally refold tubulin damaged by the stresses of motility, either by the direct action of dyneins stepping on the lattice or by bending-induced defects in the DMT wall. Such a mechanism may be especially important because intraflagellar transport machinery capable of trafficking tubulin no longer operates in mature spermatozoa[53]. Testing these hypotheses experimentally will require specific and effective in vivo small-molecule inhibitors of TRiC activity, which are not presently available. Whether the sperm-specific TRiC subunit paralogues CCT6B and CCT8L2 play a role in anchoring TRiC to RSs also remains unclear. Higher-resolution structures of RS-associated TRiC will be necessary to define the roles of sperm-specific TRiC subunits, as well as to identify the mysterious densities within the lumen and the protein/s responsible for tethering the complex to RS2. Defining precisely when TRiC became associated with the sperm axoneme during evolution—and whether it relates to fertilization mode—requires further structural studies of non-model organisms.

Our work greatly expands the catalogue of possible genetic origins of ciliopathy or infertility. The proteins we have identified as exclusive to sperm DMTs are candidates for ciliopathic disorders that impact only male fertility; these proteins may also serve as potential targets

for new male contraceptives. A lesser number of proteins, found only in non-sperm motile cilia (for example, WDR49 and NME9), are candidates for causing PCD-like phenotypes without affecting male fertility. The added complexity of sperm DMTs, in which sperm-specific proteins bind atop general ones, may also help explain why some mutations that affect conserved axonemal proteins have stronger effects on sperm tails than other types of motile cilia (Supplementary Table 7). In these scenarios, the absence of a conserved protein could trigger more extensive changes, because a greater number of proteins are dependent on the missing one. Many of these cases present with multiple morphological abnormalities of sperm flagella, suggesting that the assembly process of the sperm tail is extremely sensitive to disruption of external axonemal proteins.

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

## Methods

### Axoneme preparation

**Bovine sperm.** Frozen bovine sperm was obtained from the Utrecht University Veterinary Faculty and was prepared for cryo-EM as previously described[2]. Briefly, sperm straws were thawed by immersion in a 37 °C water bath for about 30 s. Sperm were then washed twice with Dulbecco's PBS (Sigma), counted and diluted to concentrations of approximately $0.1–0.2 \times 10^6$ cells ml$^{-1}$ in demembranation buffer (20 mM Tris-HCl pH 7.9, 132 mM sucrose, 24 mM potassium glutamate, 1 mM MgSO$_4$, 1 mM DTT and 0.1% Triton X-100). The suspension was frozen at −20 °C and thawed after 48–96 h. To expose individual DMTs, sperm were then disintegrated by the addition of ATP (Sigma) to a final concentration of 1 mM. Following 10–15-min incubation, about 4 µl of disintegrated sperm was applied to glow-discharged Quantifoil R 2/1 200-mesh holey carbon grids. Using a manual plunger (MPI), grids were blotted opposite the side of cell deposition for 5–6 s then immediately plunged into a liquid ethane–propane mix (37% ethane). Frozen grids were stored under liquid nitrogen until imaging.

**Bovine oviduct and porcine brain.** Fresh bovine oviducts and porcine brains were sourced from either Trenton Processing Center or the Division of Comparative Medicine at Washington University in St. Louis. No ethical approval or guidance was required, because organs were used from animals killed for other purposes. On receipt, oviduct and brain specimens were carefully opened and brushed, following which they were exposed to an extraction buffer (20 mM Tris pH 7.4, 50 mM NaCl, 1 mM EDTA, 7 mM β-mercaptoethanol, 10 mM CaCl$_2$, 250 mM sucrose and 0.1% CHAPS). The resultant mixture was sieved through a 300-mesh filter to remove tissue debris, followed by 2,000$g$ centrifugation for 5 min to eliminate residual tissue fragments. Subsequently, 13,000$g$ centrifugation for 30 min was performed, and the resulting pellet resuspended in resuspension buffer (RB: 30 mM HEPES pH 7.4, 5 mM MgCl$_2$, 1 mM DTT, 0.5 mM EDTA, 50 mM KCl and Roche protease inhibitor). To enhance purity, we conducted multiple rounds of centrifugation at 2,000$g$ for 2 min and 12,000$g$ for 20 min. Purified cilia were demembranated with 1% NP-40 detergent (Thermo Fisher Scientific) for 1 h at 4 °C, then subsequently centrifuged at 13,000$g$ for 20 min. The resulting pellet was resuspended in 40 µl of RB. To achieve well-separated DMTs, the sample was incubated with 1 mM ATP and 0.02 mg ml$^{-1}$ subtilisin on ice for 30 min. Samples were applied to Quantifoil R2/1 copper grids mounted in a Vitrobot Mark IV (Thermo Fisher Scientific) operated at 16 °C and 100% humidity. Following blotting for 5 s, grids were plunge-frozen in liquid ethane.

**Human oviduct.** Human oviduct samples were isolated from whole human uteri from deceased organ donors. The protocol of procurement and processing was reviewed by the Institutional Review Board of Harvard University (protocol no. IRB21-0272), which determined that tissue procurement and processing was not human subject research. No identifying information of deceased organ donors was shared in procurement or processing of tissue.

Human oviduct tissue was provided in PBS (137 mM NaCl, 2.7 mM KCl, 8 mM Na$_2$HPO$_4$ and 2 mM KH$_2$PO$_4$). For purification of human oviduct cilia, individual oviducts were cannulated near the uterotubal junction with a needle of size 18–22G, and gently flushed with 1–3 ml of of PBS using a 10-ml syringe to remove cell debris, vesicles and serous fluid before deciliation. Following a gentle 1–3 ml PBS wash, without removal of the needle, the syringe was filled with 10 ml of deciliation buffer (20 mM HEPES, 10 mM CaCl$_2$, 1 mM EDTA, 50 mM NaCl, 4% sucrose (w/v), 1 mM DTT, 1 mM dibucaine and 1× protease inhibitor cocktail (Sigma, catalogue no. S8830) per 100 ml of deciliation buffer), reconnected to the needle and the mixture flushed through the oviduct into a 50 ml conical tube. Flushing of the oviduct with 10 ml of deciliation buffer was repeated three times for each oviduct before discarding the tissue.

Conical tubes containing eluent were then spun at 900$g$ for 10 min to pellet any large debris dislodged during flushing. Following this low-speed spin, supernatant was then transferred to polycarbonate tubes and spun at 8,000$g$ for 20 min to pellet the human oviduct cilia. The pellet was then resuspended in 100 µl of of RB (30 mM HEPES, 1 mM EGTA, 4 mM MgCl$_2$, 0.1 mM EDTA, 25 mM NaCl, 1× protease inhibitor cocktail (Sigma, catalogue no. S8830) per 100 ml of RB). The resuspended pellet was then examined by negative-stain electron microscopy to determine the degree of cilia isolation. An additional round of pelleting (8,000$g$ for 20 min) was performed, and the pellet resuspended in a volume of about 20–30 µl to achieve a sample with an absorbance reading at 280 nm (A$_{280}$) of 8–10.

NP-40 at a concentration of 0.5% (v/v) was added to the purified cilia, with rotation at 4 °C for 30 min, to demembranate cilia. The resulting axonemes were then pelleted at 4 °C by centrifuging at 10,000$g$ for 20 min and resuspending in 50 µl of RB to remove any remaining NP-40 detergent. To promote DMT splaying, ATP was then added to the resuspended pellet to a concentration of 2 mM, with rotation at room temperature for 50 min. The sample was then spun at 10,000$g$ for 30 min at 4 °C and resuspended in RB, such that the A$_{280}$ value of the sample reached 10–12.

Purified human oviduct DMTs, at A$_{280}$ ranging 10–12 and at volumes of 3 µl, were placed on glow-discharged QF R2/2 grids suspended in a Vitrobot Mark IV at 4 °C and 100% humidity. Following a wait time of 10 s, the grids were blotted for 10–12 s with a force of 10–12 before being plunge-frozen in liquid ethane, then transferred to liquid nitrogen storage.

**Porcine oviduct.** Oviducts were dissected from intact porcine reproductive tracts shipped overnight, on ice, from Animal Technologies. No ethical approval or guidance was required because organs were used from pigs killed for other purposes. On receipt of the reproductive tract, the oviduct was identified at the tip of the uterine horns and dissected away from the ovary and uterine tissue before being placed in PBS. To purify porcine oviduct cilia, individual oviducts were cannulated near the uterotubal junction with a needle of gauge 18–22G. The oviduct was then gently flushed with 1–3 ml of PBS using a 10-ml syringe to remove cell debris, vesicles and serous fluid before deciliation. Without withdrawing the needle, the syringe was removed and filled with 10 ml of deciliation buffer (20 mM HEPES, 10 mM CaCl$_2$, 1 mM EDTA, 50 mM NaCl, 4% sucrose (w/v), 1 mM DTT, 1 mM dibucaine and 1× protease inhibitor cocktail (Sigma, catalogue no. S8830) per 100 ml of deciliation buffer), reconnected to the needle and the mixture flushed through the oviduct into a 50-ml conical centrifuge tube. Flushing of the oviduct with 10 ml of deciliation buffer was repeated three times for each oviduct before discarding the tissue. Centrifuge tubes containing eluent from flushing of the oviduct were then spun at 900$g$ for 10 min to pellet any large debris dislodged during flushing. Following this low-speed spin, the supernatant was then transferred to polycarbonate tubes and spun at 8,000$g$ for 20 min to pellet the cilia. The pellet was then resuspended in 100 µl of RB (30 mM HEPES, 1 mM EGTA, 4 mM MgCl$_2$, 0.1 mM EDTA, 25 mM NaCl and 1× protease inhibitor cocktail (Sigma, atalogue no. S8830) per 100 ml of RB). Having confirmed the presence of intact cilia in the resuspension by negative-stain electron microscopy, an additional round of pelleting (8,000$g$ for 20 min) was performed. Finally, the pellet was resuspended in about 20–50 µl of RB to achieve a sample with an absorbance reading of 8–15 at A$_{280}$.

Porcine oviduct cilia at A$_{280}$ ranging 8–15 were diluted with protein A-conjugated, 10-nm gold fiducials manufactured at an optical density of 10 (OD$_{10}$) (Cytodiagnostics, catalogue no. AC-10-05-15) to a final gold fiducial concentration corresponding to OD$_{5–10}$. Next, 3 µl of the mixture was placed on glow-discharged QuantiFoil R2/2 grids suspended in a Vitrobot Mark IV at 4 °C and 100% humidity. The grids were blotted 10 s following sample application for 10–12 s before being plunge-frozen in liquid ethane, then stored in liquid nitrogen before data collection.

## Mass spectrometry

**Bovine sperm.** Proteomics data of bovine sperm were previously reported[2] and are available from PRIDE, with accession no. PXD035941.

**Bovine oviduct and porcine brain.** Demembranated bovine oviduct cilia (BvOv) and porcine brain ventricle cilia (PcBV) were analysed at the Proteomics and Metabolomics Facility at the University of Nebraska-Lincoln. Cilia pellets were resuspended in RB (30 mM HEPES pH 7.4, 5 mM $MgCl_2$, 1 mM DTT, 0.5 mM EDTA and 50 mM KCl) to a final concentration of about 10 mg ml$^{-1}$. Samples were denatured at 95 °C for 10 min and run approximately 1 cm into an SDS–polyacrylamide gel electrophoresis gel. The gel was fixed for 1 h, washed and stained overnight then destained before either excision of the whole lane (BvOv) or splitting the lane into three fractions (PcBV) for further processing. All gel pieces were washed with water, reduced by the addition of dithiothreitol and alkylated with iodoacetamide before digestion with trypsin overnight at 37 °C. Peptides were dried in a speed vacuum. Digests were redissolved in 2.5% acetonitrile and 0.1% formic acid. Mass spectrometry analyses were carried out using a 2-h gradient on a $0.075 \times 250$ mm$^2$ C18 Waters CSH column feeding into an Orbitrap Eclipse mass spectrometer run in either OT–OT mode (BvOv) or OT–IT–HCD mode (PcBV).

Samples were analysed using Mascot v.2.7.0 (Matrix Science). Mascot was set up to search a common-contaminants database (cRAP_20150130.fasta, with 125 entries) and either the bovine UniProt database (37,513 sequences, downloaded 23 September 2021) or the porcine UniProt database (49,791 sequences, downloaded 20 June 2022). Mascot was searched with a fragment ion mass tolerance of 0.6 Da and parent ion tolerance of 10 ppm. Deamidation of asparagine and glutamine, oxidation of methionine and carbamidomethylation of cysteine were specified in Mascot as variable modifications.

**Human oviduct cilia.** Isolated human oviduct cilia were analysed at the Taplin Mass Spectrometry Facility at Harvard Medical School. Cilia were denatured at 95 °C using SDS and run briefly into an SDS–polyacrylamide gel. The gel piece containing ciliary proteins was excised, washed, dehydrated with acetonitrile for 10 min and dried in a speed vacuum. The gel piece was subsequently rehydrated with 50 mM ammonium bicarbonate solution containing 12.5 ng µl$^{-1}$ trypsin (Promega). After 45 min at 4 °C, the trypsin solution was removed and replaced with 50 mM ammonium bicarbonate solution to cover the gel piece, with incubation at 37 °C overnight. Peptides were later extracted by removal of the ammonium bicarbonate solution, followed by one wash with a solution containing 50% acetonitrile and 1% formic acid. Extracts were then dried in a speed vacuum for around 1 h. Dried peptide samples were reconstituted in 5–10 µl of solvent A (2.5% acetonitrile and 0.1% formic acid), then loaded onto a pre-equilibrated, nanoscale, reverse-phase high-performance liquid chromatography capillary column (100-µm inner diameter × 30-cm (approximate) length) containing 2.6 µm of C18 spherical silica beads. A gradient with increasing concentrations of solvent B (97.5% acetonitrile and 0.1% formic acid) was used to elute peptides with a Famos autosampler (LC Packings). As peptides eluted from the column, they were subjected to electrospray ionization into a Velos Orbitrap Pro ion-trap mass spectrometer (Thermo Fisher Scientific). Tandem mass spectra were acquired and analysed using Sequest (Thermo Fisher Scientific) against a protein database containing normal and reversed versions of all sequences, to determine peptide identities. Data were filtered to a peptide false discovery rate of 1–2%.

## SPA cryo-EM data collection

Details of data collection parameters are summarized in Supplementary Table 1.

**Bovine sperm.** Single-particle analysis cryo-EM data of disintegrated bovine sperm axonemes were collected using either a Talos Arctica operating at 200 kV or a Titan Krios operating at 300 kV (both Thermo Fisher Scientific). Arctica datasets were acquired in super-resolution mode on a K2 Summit direct electron detector (Gatan) with a GIF Quantum energy filter at slit width 20 eV. Krios datasets were acquired on a K3 detector (Gatan) with a BioQuantum energy filter, also with a slit width of 20 eV. Semiautomated data collection was facilitated by SerialEM[55], with data quality monitored on the fly using Warp[56]. Of the 45,431 videos processed, 34,796 were newly collected for this study and were combined with 10,635 from our previously reported dataset[2].

**Human oviduct cilia.** SPA cryo-EM data of disintegrated human oviductal axonemes were collected on a Titan Krios microscope at Harvard Medical School. Videos were recorded on a K3 camera with a BioQuantum energy filter at slit width 20 eV. A $2 \times 2$ beam tilt pattern with three images per hole was utilized to increase the data collected per stage movement. Each stage position was selected manually to avoid areas with contamination or few DMTs. Images were collected semiautomatically using SerialEM[55].

**Bovine oviduct cilia and porcine ependymal cilia.** SPA cryo-EM data of DMTs from bovine oviduct cilia (4,716 videos) and porcine ependymal cilia (7,051 videos) were collected using Titan Krios microscopes at Case Western Reserve University. Images were collected semiautomatically using SerialEM[55].

## Cryo-ET data collection

Tilt series were collected on vitrified porcine oviductal cilia using a Titan Krios microscope, operated at a nominal magnification of ×53,000 (corresponding to a pixel size of 1.68 Å) and equipped with K3 camera and a BioQuantum energy filter at slit width 20 eV. SerialEM[55] was used for data collection. Targets were selected by identifying cilia in medium-magnification montages. Automatic tilt series were then collected on these targets using a dose-symmetric scheme[57], collecting from −54 to +54° with 3° between tilts and targeting a total dose of 110 e/Å$^2$. Data acquisition parameters are summarized in Supplementary Table 1.

## SPA cryo-EM data processing

Single-particle data of axonemal DMTs from all cilium types were processed using the same workflow, to ensure consistency between results. All maps reported here represent consensus averages of all nine DMTs, because information about their relative positions is lost during axoneme splaying.

Video frames were drift corrected and dose weighted using patch motion correction in cryoSPARC[58]. Contrast transfer function (CTF) parameters were estimated using patch CTF estimation in cryoSPARC. DMTs were automatically picked using filament tracer in cryoSPARC. Next, DMT particles were extracted along filament traces using overlapping boxes of 8-nm step size. DMT particles (256-pixel box size, 2× binning) were subjected to two rounds of two-dimensional classification to remove junk and off-centred particles. Good DMT particles then underwent structural refinement using Homogeneous Refinement (New) in cryoSPARC.

Next, these DMT particles and their alignment parameters were exported to FREALIGN v.9.11 and underwent local refinement. In this step we used customized scripts to minimize alignment errors, based on the geometric relationship of neighbouring DMT particles. The particle set with improved alignment parameters was imported back to cryoSPARC for one round of local refinement, followed by tubulin signal subtraction.

For separation of 48-nm repeat from 8-nm particles, we performed three-dimensional classification of tubulin-subtracted DMT particles in Relion 3.1 (ref. 59) using a soft-edged mask covering MIPs near

protofilaments A08–A13. A similar strategy was used to further split 48-nm particles into two sets of 96-nm particles, using a soft-edged mask covering an external region near protofilaments A01–A03. The coordinates of 48 and 96-nm DMT particles were imported back to cryoSPARC, re-extracted at 512-pixel box size (no binning) and subjected to one round of local refinement followed by local CTF refinement. This step produced consensus 48 and 96-nm DMT maps. Due to computational constraints, we used a box size of 512 pixels (666 Å) for three-dimensional reconstruction. As a result, we used four different reconstruction boxes whose centres were 24 nm apart (positions 1–4) to cover the 96-nm repeat length (Supplementary Fig. 1).

For improvement of DMT local resolution, we performed focused refinements in cryoSPARC using a set of cylindrical masks as described in ref. 2. These masks divide the DMT into 39 subregions. In regard to external 96-nm features such as RSs and IDAs, we used a similar divide-and-conquer strategy. We first shifted the centre of the nearest reconstruction box (among positions 1–4; Supplementary Fig. 1) to the feature of interest using customized scripts, and then performed three-dimensional classification and focused refinement for the local region (Supplementary Figs. 2–7). In most cases, three-dimensional classification of tubulin-subtracted particles produced better results, and the masks used for three-dimensional classification and focused refinement were adjusted iteratively until no further improvement was observed. For external complexes, a rough total of 20 local regions was refined for each cilium type, with resolution estimates provided in Supplementary Table 2.

To generate a composite map for model building and refinement, we prepared a large rectangular box (600 × 640 × 1,024 pixels) covering the entire length of the 96-nm repeat, by stitching together the two halves of the 96-nm DMT maps. We refer to this rectangular box as the 'big map'. All reconstructions for local regions were sharpened using deepEMhancer[60], which produced a consistent grey level across various maps. Sharpened maps were multiplied by their respective masks and aligned to the big map using the fit in map command in Chimera[61]. Aligned maps were resampled onto the grid of the big map, and merged using the vop resample and vop maximum commands in Chimera.

## Cryo-ET data processing

Videos of ten frames, recorded at each tilt angle, were motion corrected and coarsely aligned into a tilt series of single micrographs using alignframes from IMOD[62]. Motion-corrected micrographs were manually inspected and removed from the tilt series using etomo if uncorrected drift was observed. Following automatic detection with IMOD, each fiducial position was manually inspected to ensure correct fiducial tracking through the tilt series. The CTF was fit using IMOD's Ctfplotter, and tomograms were generated in IMOD using back projection.

Within each tomogram, DMTs were manually traced using IMOD's graphical user interface. Particle positions were then placed every 8 nm along the traced DMTs. Initial translation and angular alignment searches at bin 8 (pixel size 18.24 Å, box 200 pixels) for each particle position were then performed in PEET[6] using a 96-nm reconstruction of the *T. thermophila* DMT (EMD-9023)[54] as an initial reference, low-pass filtered to 50 Å. Translational search distances allowed for alignment on the nearest 24-nm repeat. Following initial alignment in PEET, particle positions and Euler angles were then imported to RELION 4.0.1 (ref. 63) for classification, and for additional refinement of alignment and CTF parameters. In RELION, pseudosubtomograms were extracted at each particle position with 8× binning and subjected to a round of local refinement. Subsequently, three-dimensional classification was performed with a cylindrical mask covering the RSs and inner dynein arms (Extended Data Fig. 1e). Classification on density within this cylinder led to separation of 96-nm registers of the DMT, one of which was selected for subsequent reconstruction. Pseudosubtomograms of these particles were then extracted at bins 4, 2 and 1, with local

refinement performed before each decrement in bin size. At bin size 1 (pixel size 1.68 Å, box 220 pixels), refinement of CTF parameters and frame alignment was performed to improve the quality of the reconstruction. Because the maximally achieved resolution at bin 1 (8.4 Å, based on the Fourier shell correlation 0.143 criterion) was greater than Nyquist frequency at bin 2 (6.64 Å), subsequent maps of the DMT were aligned at a bin size of 2 or 4.

For reconstruction of a 96-nm map of the porcine oviduct DMT at bin 2 (pixel size 3.36 Å, box 340 pixels), focused refinement was performed on the central 60-nm portion of the DMT. Particles were then shifted along the long axis of the DMT by about 24 nm, followed by a further focused refinement. This was repeated until four overlapping porcine oviduct DMT maps were obtained, shifted by roughly 24 nm and with resolution of 9.2–10.2 Å. A composite 96-nm map was then constructed using vop maximum in ChimeraX[64], to record maximum voxel density of the overlapped maps. A map of the 96-nm repeat, including axonemal complexes, was performed at bin 4 (pixel size 6.72 Å, box 340 pixels), with an estimated resolution of 23 Å based on the Fourier shell correlation 0.5 criterion. The resulting map represents a consensus average of all nine DMTs. Note that cryo-ET data were processed independently of SPA data and that the subtomogram average of porcine oviduct DMTs was not used in SPA processing of other axoneme types.

## Model building and refinement

Models of the 96-nm axonemal repeat from bovine sperm were built based on available structures of human respiratory axonemes (PDB 8J07)[1] and of bovine sperm DMTs (PDB 8OTZ)[2]. Human proteins were replaced with either predictions from AlphaFold2 (ref. 65) or homology models from SWISS-MODEL[66] using the most similar *B. taurus* sequences from UniProt or NCBI.

Densities unaccounted for by these models were assigned through either sequence- or structure-based approaches. The strategy applied for each newly assigned protein, along with supporting evidence from the literature, is summarized in Supplementary Figs. 8–33. For regions with well-resolved side-chain densities, backbone traces were built either manually with Coot[67] or automatically with ModelAngelo[68]. Either findMySequence[69] or ModelAngelo was then used to estimate side-chain probabilities, and to find the best-matching sequence from our bovine sperm proteome. For densities at intermediate resolution (around 5 Å), we applied one of three structure-based approaches, either (1) manual tracing of helices in Coot, followed by querying Alpha-Fold2 databases using the DALI server[70], deepTracerID[71] or FoldSeek[72]; (2) automatic fitting of AlphaFold predictions into segmented density using the colores algorithm[73] in the Situs package[74], followed by ranking based on cross-correlation scores and manual inspection of top hits[75]; or (3) using a density-based fold-recognition algorithm based on MOLREP–BALBES, followed by manual inspection of top hits[76]. To increase confidence in assigning unique proteins to unknown densities, reverse searches were performed using DALI or FoldSeek, and AlphaFold2 predictions of candidate proteins as queries. If multiple proteins could fit equally well, these alternatives are noted in Supplementary Figs. 8–33.

AlphaFold2 predictions of newly identified proteins or protein subcomplexes were fit into the density maps using ChimeraX[64], followed by manual adjustment in Coot and molecular dynamics flexible fitting with Namdinator[77]. Individual PDB files were merged and given unique chain IDs in ChimeraX, then real-space refined in Phenix[78] using a non-bonded weight of 500. Due to the size of the model, the 96-nm repeat was split into two halves, each refined independently. Model statistics are summarized in Supplementary Table 5.

## Reporting summary

Further information on research design is available in the Nature Portfolio Reporting Summary linked to this article.

## Data availability

Composite cryo-EM maps of the 96-nm repeat of axonemal DMTs from bovine sperm, bovine oviductal cilia, human oviductal cilia and porcine brain ventricle cilia have been deposited to EMDB with codes EMD-50664, EMD-45783, EMD-45785 and EMD-45784, respectively. Local refinements for bovine sperm have been deposited to EMDB with codes EMD-50866 and EMD-50886; for bovine oviduct with codes EMD-45683 and EMD-45697; for human oviduct with codes EMD-45714, EMD-45725 and EMD-45790; and for porcine brain ventricle with codes EMD-45699 and EMD-45713. Subtomogram averages of the 96-nm repeat of axonemal DMTs from porcine oviductal cilia have been deposited with codes EMD-45677 and EMD-45680. Cryo-EM maps of the 48-nm DMT repeat from bovine oviductal cilia and porcine brain ventricle cilia have been deposited with codes EMD-45801 and EMD-45802, respectively. The atomic model of the 96-nm repeat of bovine sperm DMT has been deposited to PDB with accession code PDB 9FQR. Atomic models of the 48-nm DMT repeat from bovine oviductal cilia and porcine brain ventricle cilia have been deposited to PDB with accession codes PDB 9CPB and PDB 9CPC, respectively. Previously reported atomic models of the 96-nm repeat from human respiratory cilia, and of the 48-nm repeat from bovine sperm were used as initial models and are available with PDB accession codes PDB 8J07 and PDB 8OTZ, respectively. Proteomics data from bovine oviductal cilia, human oviductal cilia and porcine brain ventricle cilia are available in Supplementary Table 4.

## Code availability

Custom scripts used in this study are publicly available at https://github.com/rui--zhang/Doublet.

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

**Acknowledgements** We thank A. Rijneveld at the Utrecht University Faculty of Veterinary Medicine for providing bovine sperm straws; Trenton Processing Center (IL, USA) and T. Pavek at the Division of Comparative Medicine at Washington University in St. Louis for providing bovine fallopian tubes and porcine brains; and K. L. McKinley for help in sourcing human oviducts. We thank S. Roet and M. Vanevic for computational support at Utrecht University, and acknowledge S. C. Howes, M. Bergmeijer, C. Schneijdenberg and J. Meeldijk for management and maintenance of the Utrecht University Electron Microscopy Centre. We thank M. C. Roelofs for help with setting up the SITUS/colores pipeline. We thank K. Adelman for advice on analysis of single-cell RNA sequencing data, and R. Tomaino for mass spectrometry analysis. We thank M. Naldrett and S. Alvarez at the Proteomics and Metabolomics Facility at the University of Nebraska-Lincoln for proteomics analysis of bovine oviduct and porcine brain ventricle cilia. Cryo-EM data of bovine sperm were collected at the Utrecht University Electron Microscopy Centre and at the Netherlands Centre for Electron Nanoscopy. Cryo-EM data of human and porcine oviductal cilia were collected at The Harvard Cryo-EM Center for Structural Biology at Harvard Medical School. Cryo-EM data of bovine oviduct and porcine brain ventricle cilia were imaged at Case Western Reserve University, with the assistance of K. Li. This work benefited from the Netherlands Electron Microscopy Infrastructure (project no. 184.034.014) of the National Roadmap for Large-Scale Research Infrastructure of the Dutch Research Council. We thank the Department of Biochemistry and Molecular Biophysics at Washington University in St. Louis for the seed grants given to C.S. and J.Z. A.B. was supported by NIGMS grant nos. R01GM141109 and R01GM143183 and by the Smith Family Foundation. T.Z.-B.-M. was funded by NWO ENW-XL grant no. OCENW.XL21.XL21.048. R.Z. was funded by NIGMS grant no. R01GM138854.

**Author contributions** M.R.L. prepared bovine sperm samples and collected data for single-particle analysis with W.E.M.N. C.S., J.Z. and Q.N. prepared bovine oviduct and porcine brain cilia samples. C.S. collected data for single-particle analysis with W.H. J.R.A. prepared human oviduct cilia samples for single-particle analysis, collected tilt series for porcine oviduct cilia and analysed cryo-ET data. M.R.L., C.S., J.R.A. and R.Z. performed single-particle data processing. M.R.L., J.Z., A.B., T.Z.-B.-M and R.Z. identified proteins and built atomic models. All authors provided critical feedback and helped to formulate the research, analysis and writing of the paper.

**Competing interests** The authors declare no competing interests.

**Additional information**
**Correspondence and requests for materials** should be addressed to Alan Brown, Tzviya Zeev-Ben-Mordehai or Rui Zhang.

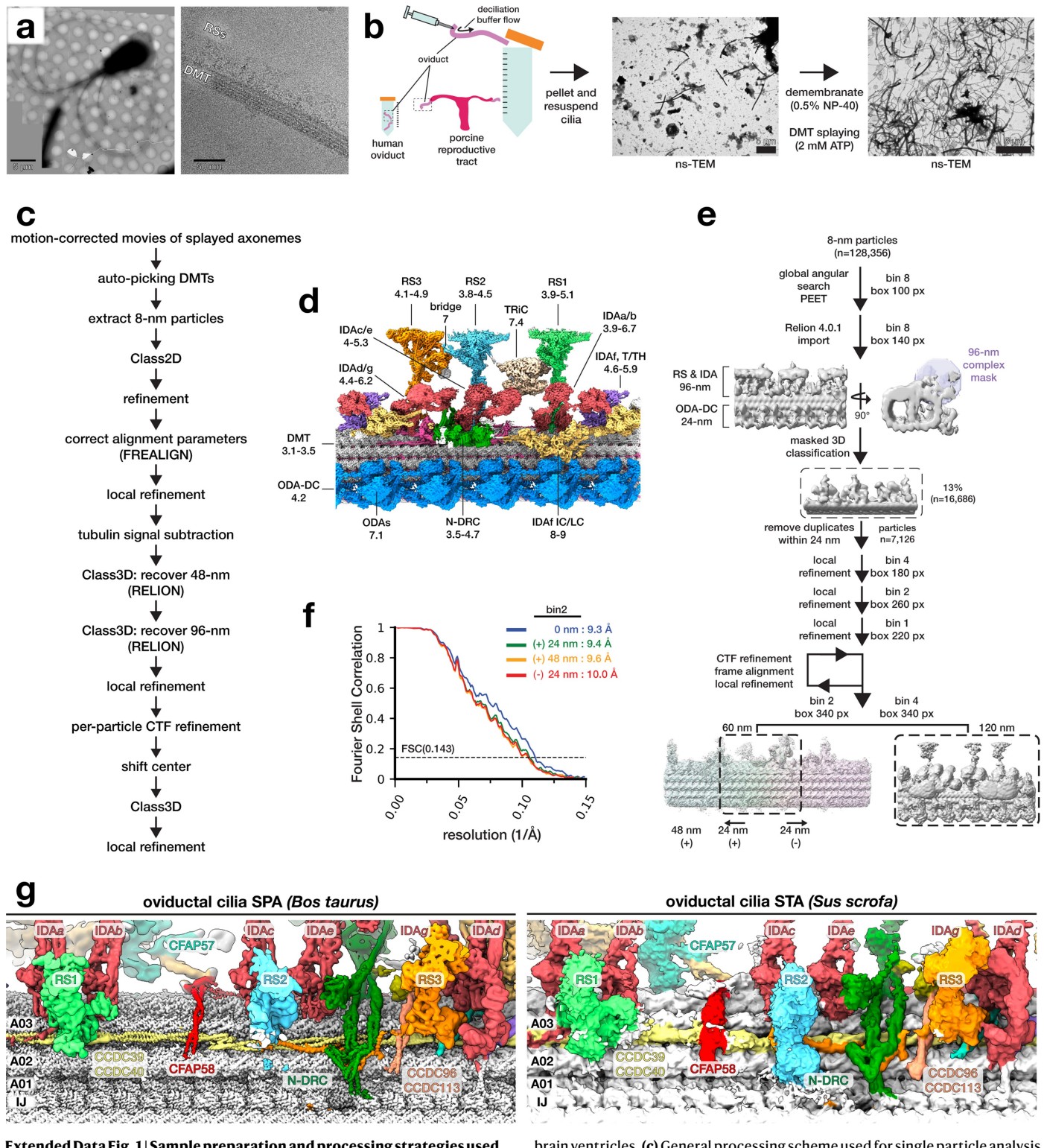

**Extended Data Fig. 1 | Sample preparation and processing strategies used to resolve the 96-nm repeat of axonemal doublet microtubules (DMTs) from different mammalian motile cilia. (a)** Stitched low-magnification micrographs of a disintegrated bovine sperm cell (left) and corresponding high-magnification image of a DMT with bound radial spokes (RSs) (right). A total of 45,431 such micrographs were processed for this study. **(b)** Schematic of the isolation of human and porcine oviduct cilia using a syringe inserted into the oviduct. Human cilia were splayed into individual DMTs after demembranating and incubating with ATP to promote DMT-DMT sliding. A similar workflow was used to prepare DMTs from bovine oviducts and porcine

brain ventricles. **(c)** General processing scheme used for single particle analysis (SPA) of DMTs from bovine sperm, bovine/human oviductal cilia, and porcine brain ventricle cilia. All steps were performed in cryoSPARC unless otherwise stated. See Methods and Supplementary Figs. 1–7 for details. **(d)** Cryo-EM map of the bovine sperm 96-nm repeat with local resolution ranges for individual axonemal complexes indicated. **(e)** Porcine oviduct cilia subtomogram averaging (STA) workflow. See Methods for details. **(f)** Fourier shell correlation (FSC) curves for the four segments used to reconstruct the porcine DMT at binning factor 2 by STA. **(g)** Comparison of oviduct cilia reconstructions obtained by SPA and STA.

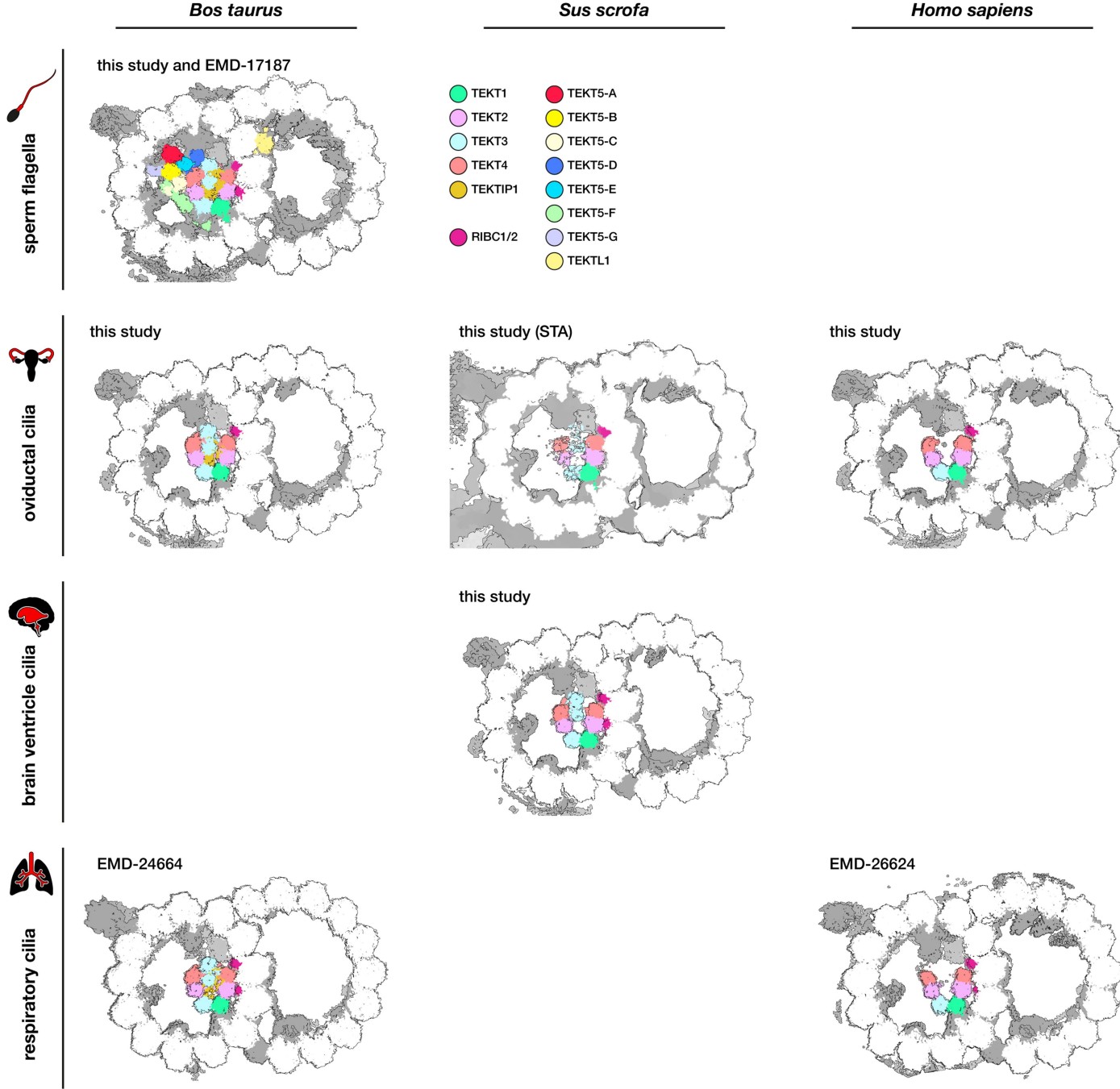

**Extended Data Fig. 2 | Differences in the tektin bundle across mammalian motile cilia.** Cross-sections of doublet microtubules from the indicated cilia types and species. Tektins (TEKT1-5), Tektin-like protein 1 (TEKTL1), tektin-interacting protein 1 (TEKTIP1), and RIBC1/2 are colored according to the legend. Other microtubule-binding proteins are in grey and tubulin is in white. Structures were determined by single-particle analysis (SPA) except for oviductal cilia from *Sus scrofa*, which was determined by subtomogram averaging (STA).

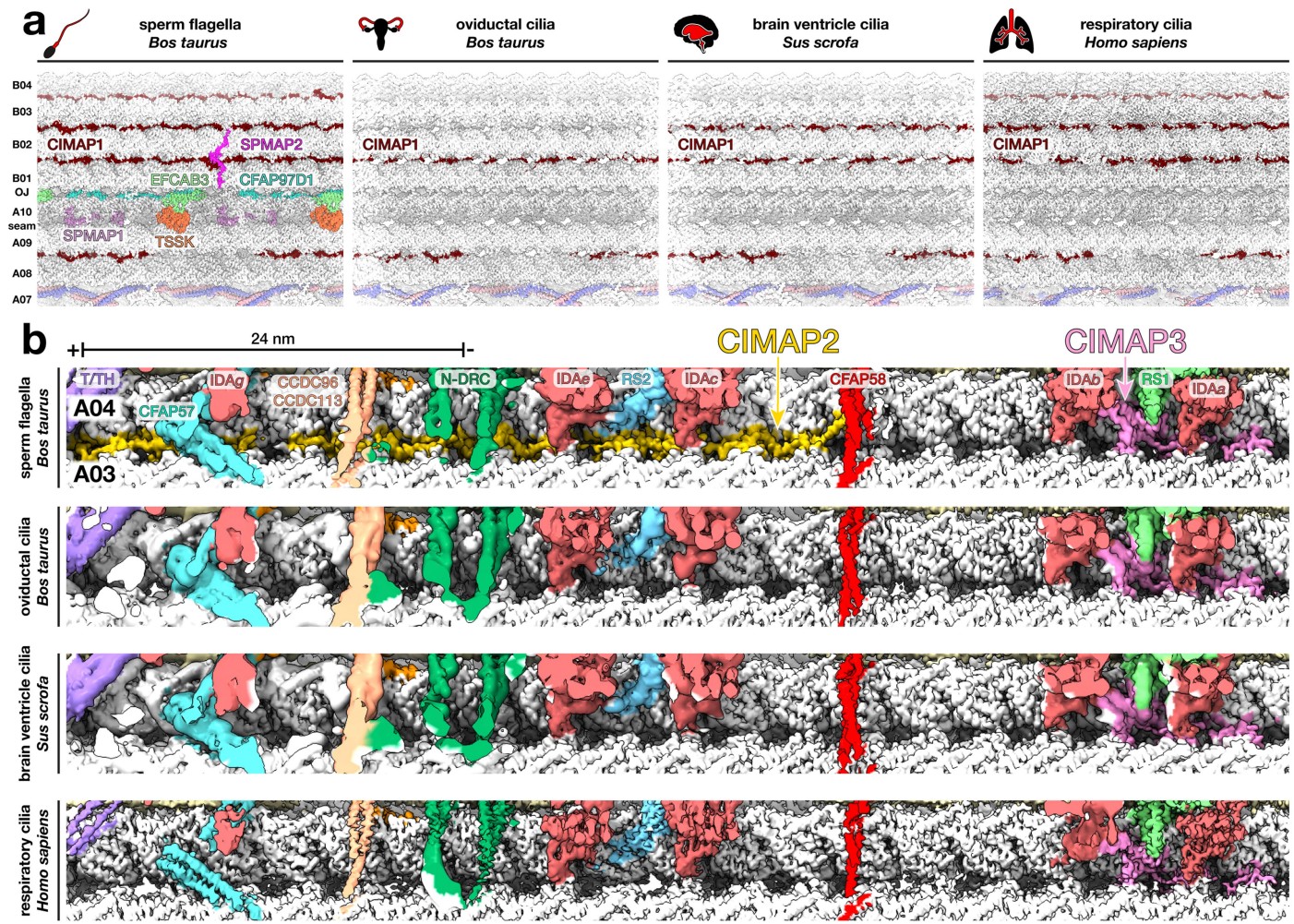

**Extended Data Fig. 3 | Comparison of ciliary microtubule associated proteins (CIMAPs) across mammalian motile cilia. (a)** CIMAP1 family proteins are present in all motile cilia examined, but sperm have several additional MAPs including SPMAP1/2, CFAP97D1, EFCAB3, and TSSK. **(b)** CIMAP2 (only detected in sperm) and CIMAP3 (in all cell types) interact with external axonemal complexes.

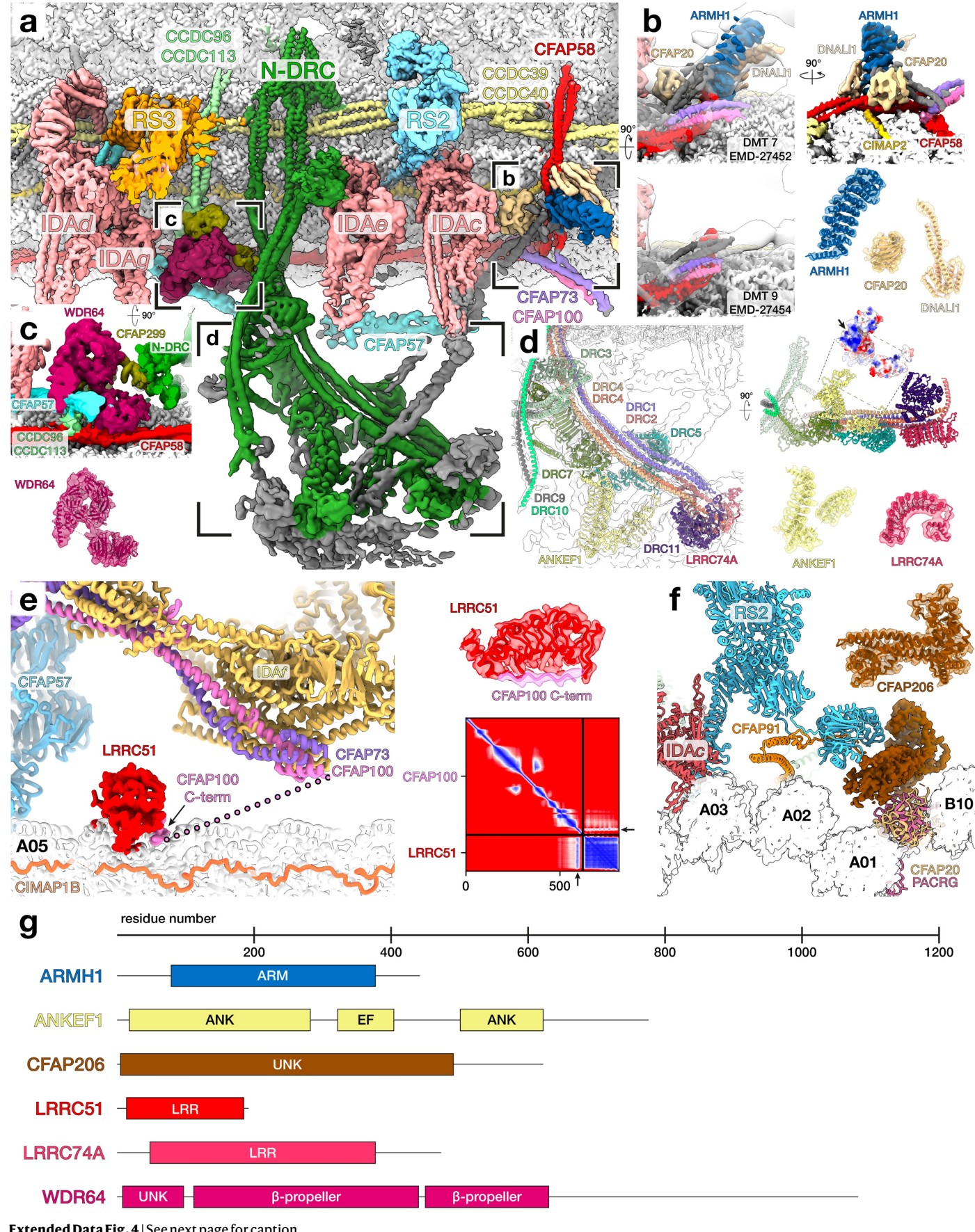

**Extended Data Fig. 4** | See next page for caption.

**Extended Data Fig. 4 | Binding sites and interactions of newly identified proteins associated with the nexin-dynein regulatory complex (N-DRC), inner dynein arm *f* (IDA*f*), and radial spoke 2 (RS2). (a)** Top-down view of the bovine sperm DMT surface around RS2, N-DRC, and RS3. **(b)** A complex of ARMH1, CFAP20, and a DNALI1 dimer sits atop CFAP58. This complex represents new binding sites for CFAP20, which is also found at the inner junction and in the central apparatus[79], and DNALI1, which is also associated with single-headed IDAs. ARMH1 is linked to the N-DRC via unidentified coiled-coils (grey). Comparison with doublet-specific subtomogram averages from mouse sperm[10] (translucent white map) suggests that the ARMH1 complex is asymmetrically distributed around the axoneme. **(c)** WDR64 binds on top of the CCDC96/113 coiled-coil, where it also interacts with CFAP299, CFAP57, and CFAP58. WDR64 is sperm-specific but may be replaced by WDR49 in other cell types. **(d)** ANKEF1 and LRRC74A are newly assigned components of the N-DRC distal lobe. ANKEF1 presents a positively charged surface that could bind glutamylated tubulin on the neighboring DMT. **(e)** LRRC51, which is present in all cilia types studied, binds the DMT surface directly underneath IDA*f*, between protofilaments A05 and A06 (left panel). AlphaFold Multimer predictions suggest that the elongated density beside LRRC51 could correspond to the C-terminus of CFAP100 (right panel). **(f)** CFAP206 bridges the inner junction and the base of RS2, thus contributing to the docking of this radial spoke. **(g)** Domain organization for ARMH1, ANKEF1, CFAP206, LRRC51, LRRC74A, and WDR64.

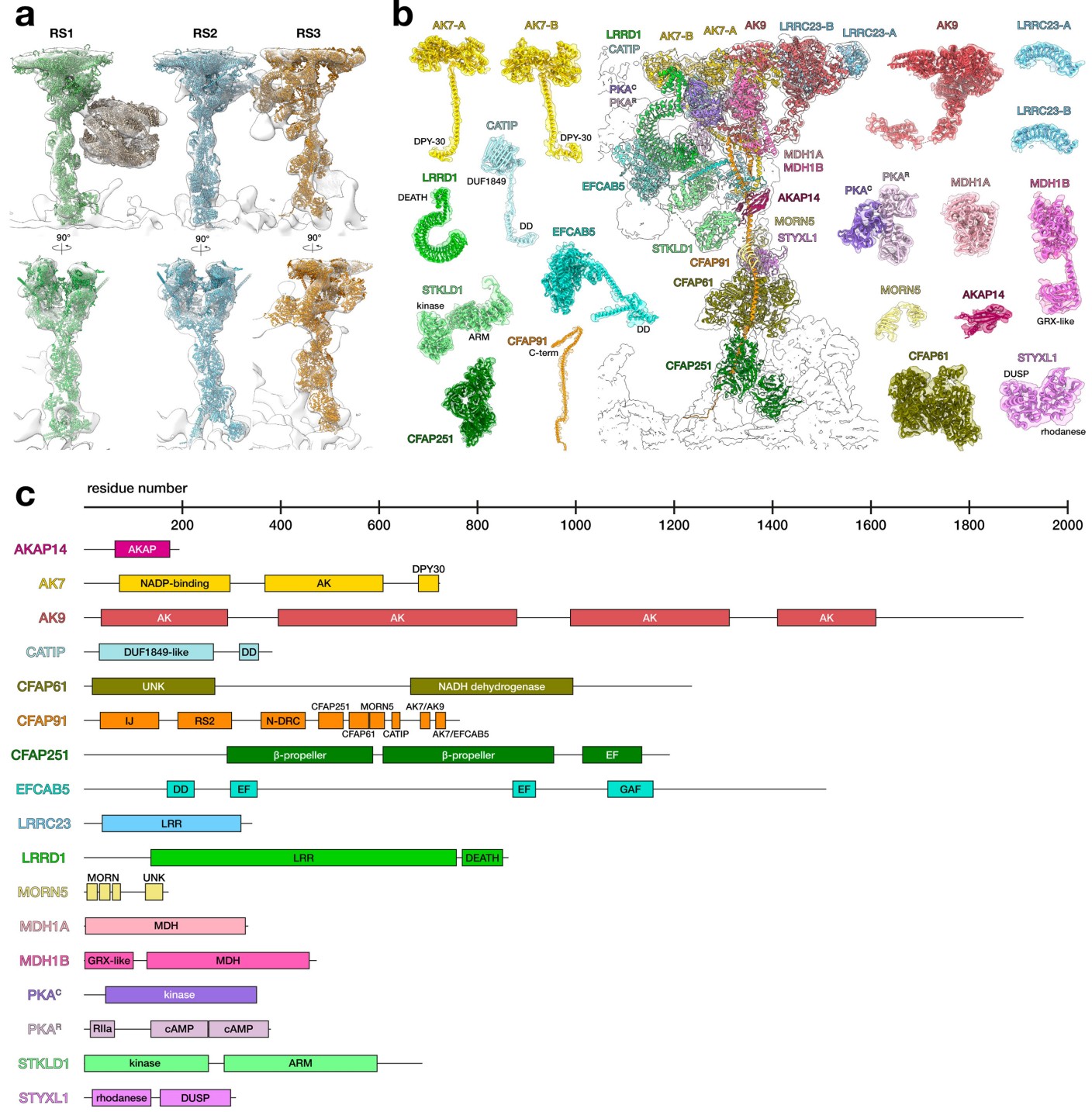

**Extended Data Fig. 5 | Structure and composition of sperm radial spoke 3 (RS3). (a)** Atomic models of RS1, RS2, RS3, and RS-tethered TRiC fit into the consensus subtomogram average of the 96-nm repeat from mouse sperm (EMD-27444, Ref. 10). **(b)** Model-in-map fits and **(c)** domain organization for RS3 proteins assigned in bovine sperm. For CFAP91, the schematic instead indicates regions interacting with other RS3 proteins or axonemal complexes.

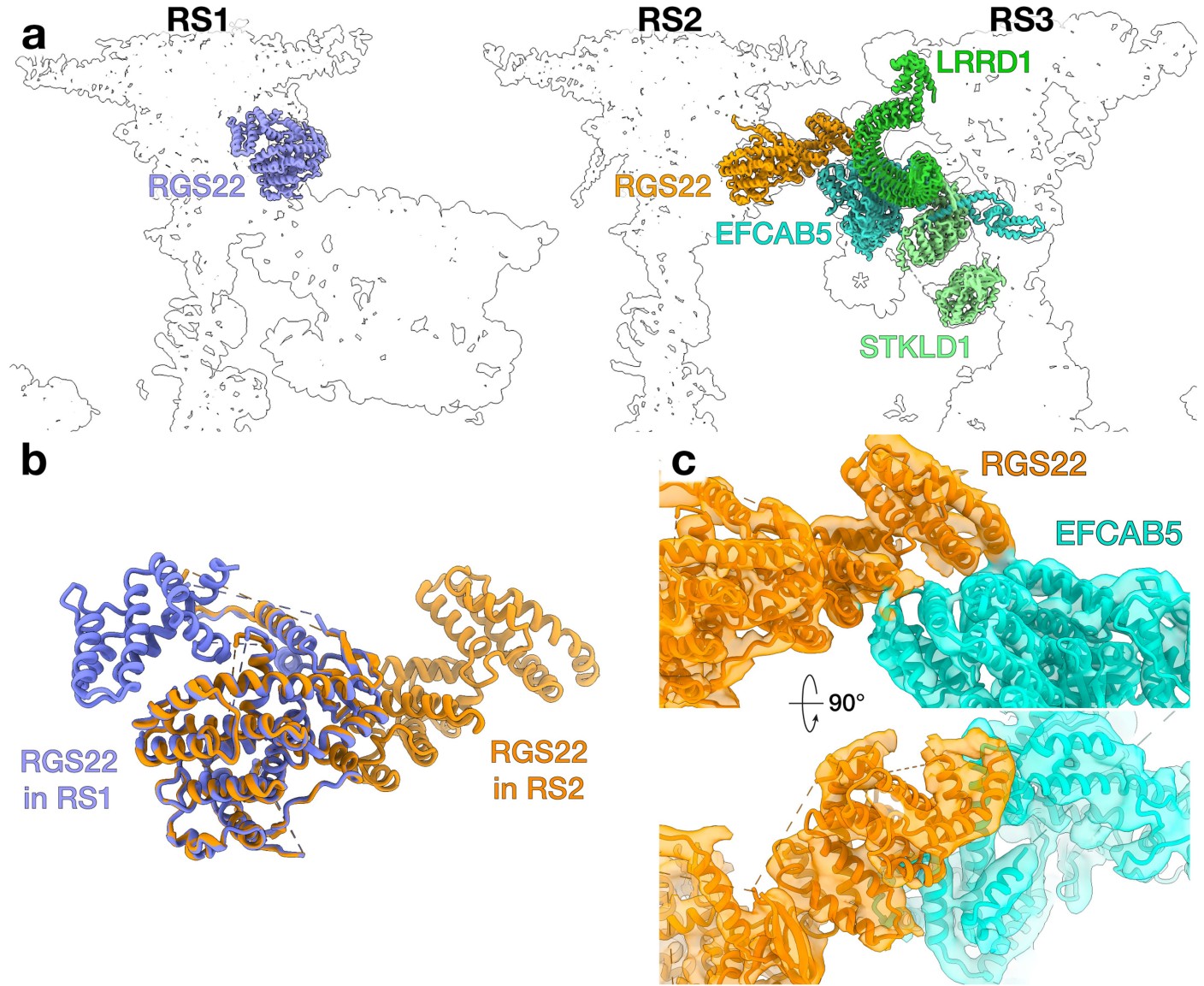

**Extended Data Fig. 6 | Structure of the sperm-specific RS2-RS3 bridge.**
**(a)** The bridge between RS2 and RS3 is formed by the interaction of RGS22, a conserved ciliary protein, with EFCAB5, a sperm-specific protein. The sperm-specific RS3 proteins LRRD1 and STKLD1 interact with EFCAB5 to further support the bridge, along with an unassigned globular density (asterisk). **(b)** Superposition of RGS22 from RS1 and RS2 showing a large conformational change, potentially induced by binding EFCAB5. **(c)** Cryo-EM density and corresponding atomic models are shown for the interaction site between RGS22 and EFCAB5.

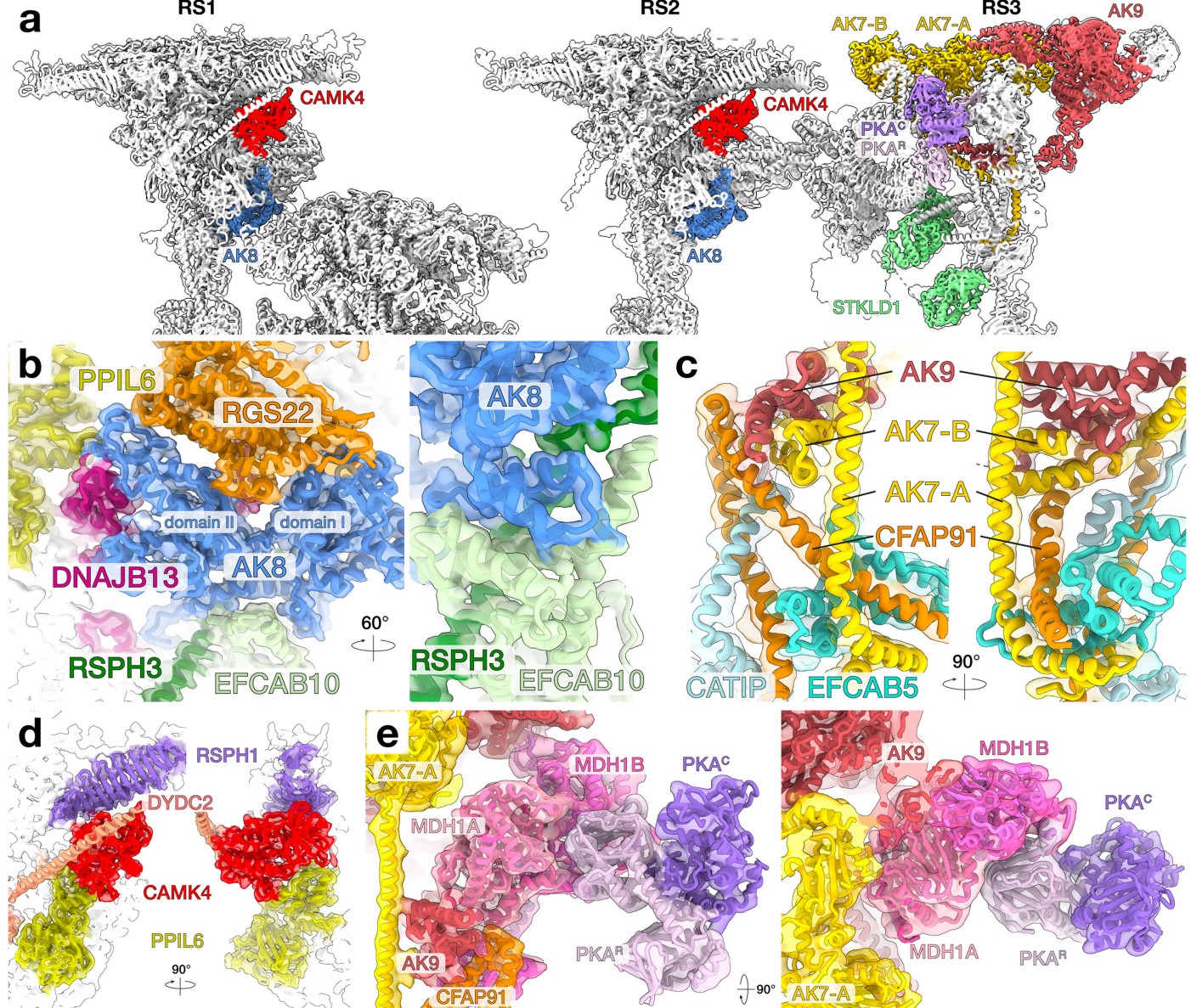

**Extended Data Fig. 7 | Structural bases of kinase anchoring to radial spokes.**
**(a)** Overview showing the positions of kinases (colored) in the radial spokes of the bovine sperm DMT. **(b)** Interactions of adenylate kinase 8 (AK8) with PPIL6 and DNAJB13 stabilize the open conformation of catalytic domain II (left). AK8 is anchored in the neck of RS1 and RS2 via its N-terminal dimerization/docking domain, which interacts with EFCAB10 and packs against RSPH3 (right). **(c)** AK7 and AK9 are anchored to RS3 via their C-terminal DPY30 domains. The DPY30 domain of AK7-A dimerizes with the DPY30 motif in EFCAB5, while the DPY30 domain of AK7-B interacts with AK9; both pairs of proteins then dock onto CFAP91. **(d)** CAMK4 interacts with RSPH1, DYDC2, and PPIL6 in the heads of RS1 and RS2. See also the Supplementary Fig. 14. **(e)** The ciliary PKA holoenzyme is a heterodimer of a catalytic (PKA$^C$) and regulatory (PKA$^R$) subunit. PKA$^R$ interacts with a malate dehydrogenase dimer (MDH1A/MDH1B) in the center of the RS3 head.

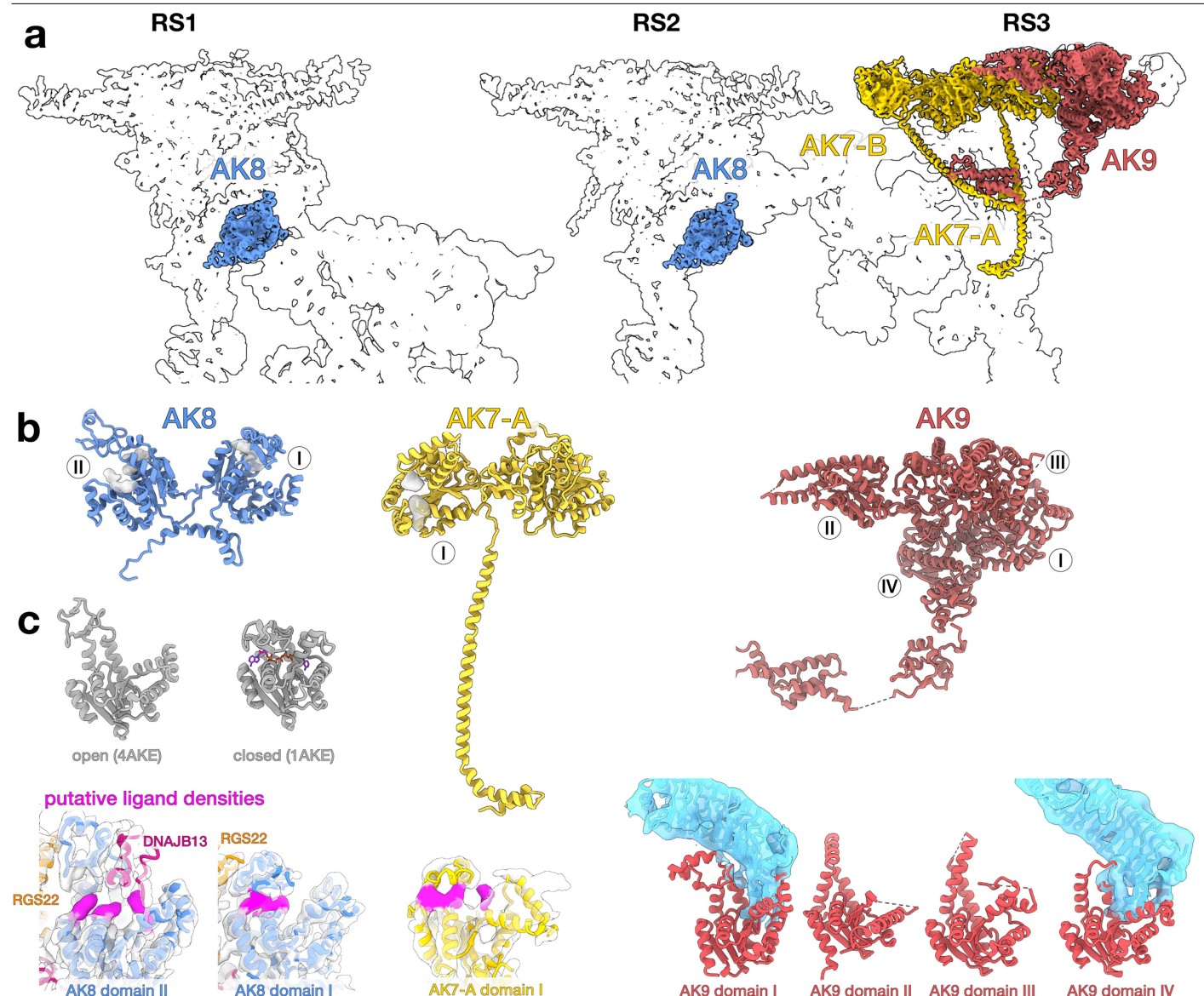

**Extended Data Fig. 8 | Structures and conformations of radial spoke-anchored adenylate kinases (AKs). (a)** RS1 and RS2 each have one copy of AK8, while RS3 has two copies of AK7 and one copy of AK9. **(b)** Models of axonemal AKs with catalytic domains labelled. **(c)** Catalytic domains of axonemal AKs are observed in either open (PDB 4AKE)[23] or closed (PDB 1AKE)[22] conformations.

Densities consistent with small molecules in the putative nucleotide-binding pockets are shown in pink. These densities were colored by aligning PDB 1AKE to the relevant adenylate kinase domain and coloring density within 3 Å of the ligand atoms.

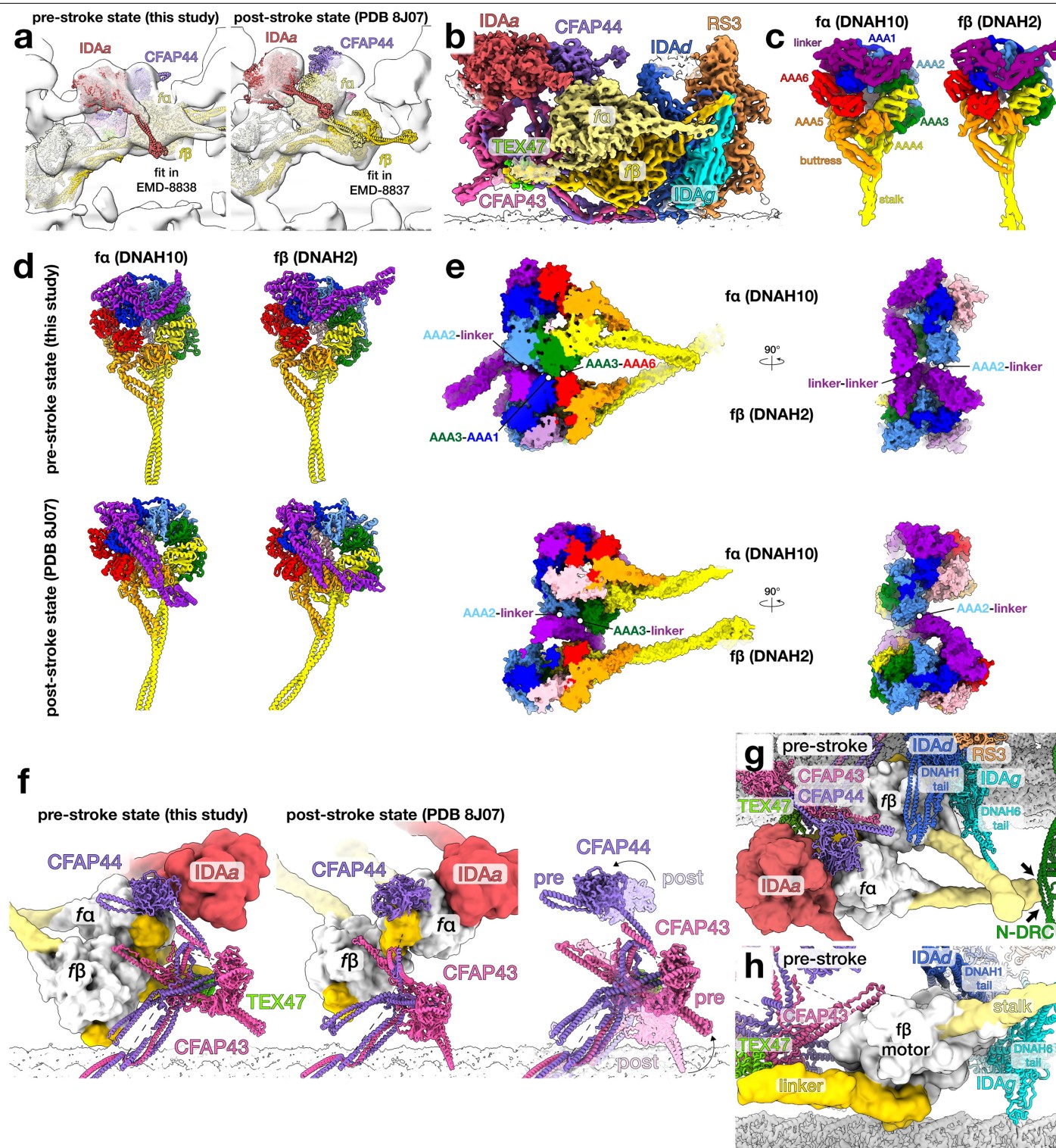

**Extended Data Fig. 9 | Conformational changes in inner dynein arm *f* (IDA*f*) and the tether/tetherhead (T/TH) complex. (a)** Comparison of our atomic models with subtomogram averages of IDA*f* in the pre- and post-stroke states, obtained from rapidly frozen live, swimming sea urchin sperm (Lin & Nicastro 2018). **(b)** Composite map of the motor domains of IDA*f* (*f*α and *f*β), IDA*a*, and the T/TH complex (CFAP43, CFAP44, and the sperm-specific TEX47). **(c)** Cryo-EM density for the motor domains of IDA*f* (DNAH10 and DNAH2). The linker is colored purple and individual AAA domains are colored from blue to red. **(d)** Changes in the conformation of the linker and stalk domains of the dynein

heavy chains (upper panel: pre-stroke state, lower panel: post-stroke state). **(e)** Relative orientations of the two motors of IDA*f* change from perpendicular arrangement in the pre-stroke state (upper panels) to a parallel arrangement in the post-stroke state (lower panels). **(f)** Conformational changes in the T/TH complex, consisting of CFAP43, CFAP44, and the sperm-specific TEX47. **(g)** Interaction of the motor domain of *f*α with CFAP44 and the motor domain of IDA*a* via CFAP44. The microtubule-binding domain of *f*α interacts with the N-DRC (arrows). **(h)** The motor domain of *f*β interacts with CFAP43 and IDA*d*, while its linker interacts with TEX47 and its stalk with IDA*g*.

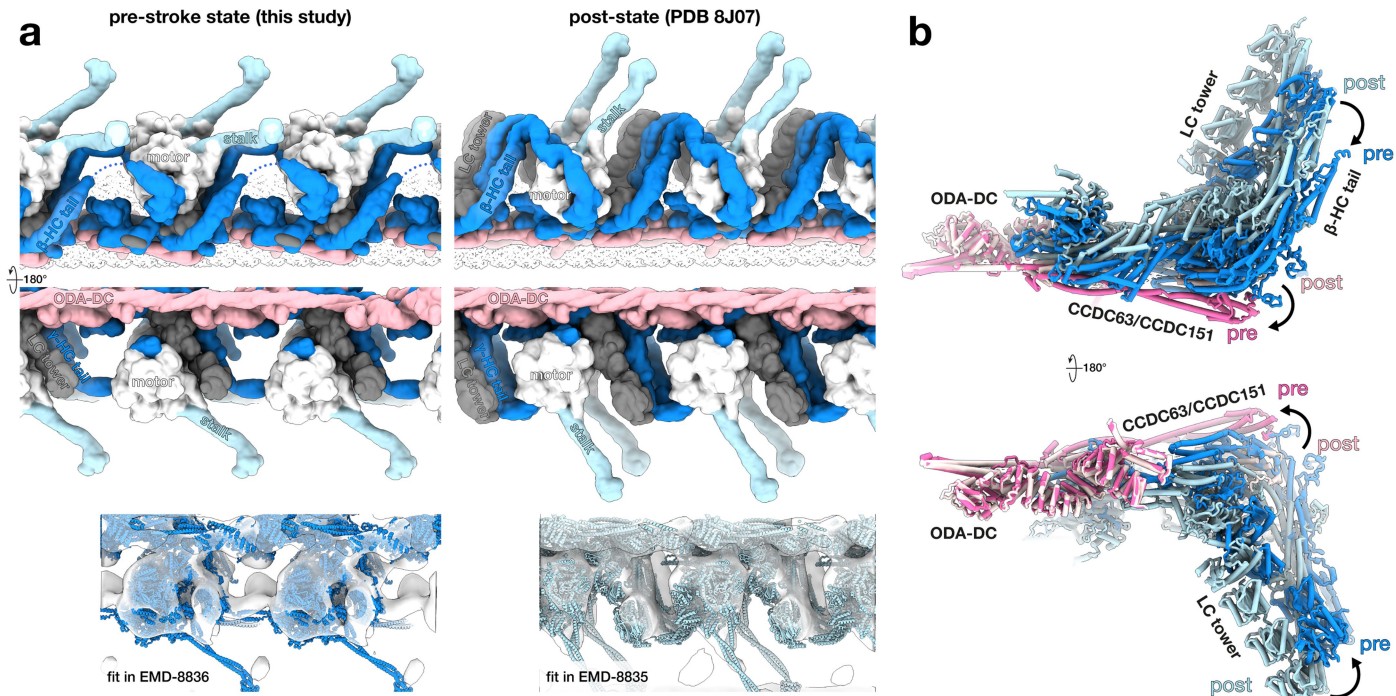

**Extended Data Fig. 10 | Conformational changes in outer dynein arms (ODAs).**
**(a)** Top panels show molecular surface representations of ODAs in conformations resembling the pre-stroke state (this study) and post-stroke state (PDB 8J07)[1]. Bottom panel shows comparison of our atomic models with *in situ* subtomogram averages of ODAs in the pre- and post-stroke states from rapidly frozen actively swimming sea urchin sperm[40]. **(b)** Overlaid atomic models showing how transition from post- to pre-stroke state involves rotation of the distal CCDC63/CCDC151 coiled-coil, the IC/LC block, and the linkers of dynein heavy chains in order to accommodate the ~8-nm proximal shift of the motor domains from the neighboring dyneins.

# Reporting Summary

## Statistics

For all statistical analyses, confirm that the following items are present in the figure legend, table legend, main text, or Methods section.

| n/a | Confirmed | |
|---|---|---|
| ☒ | ☐ | The exact sample size (*n*) for each experimental group/condition, given as a discrete number and unit of measurement |
| ☒ | ☐ | A statement on whether measurements were taken from distinct samples or whether the same sample was measured repeatedly |
| ☒ | ☐ | The statistical test(s) used AND whether they are one- or two-sided *Only common tests should be described solely by name; describe more complex techniques in the Methods section.* |
| ☒ | ☐ | A description of all covariates tested |
| ☒ | ☐ | A description of any assumptions or corrections, such as tests of normality and adjustment for multiple comparisons |
| ☒ | ☐ | A full description of the statistical parameters including central tendency (e.g. means) or other basic estimates (e.g. regression coefficient) AND variation (e.g. standard deviation) or associated estimates of uncertainty (e.g. confidence intervals) |
| ☒ | ☐ | For null hypothesis testing, the test statistic (e.g. *F*, *t*, *r*) with confidence intervals, effect sizes, degrees of freedom and *P* value noted *Give P values as exact values whenever suitable.* |
| ☒ | ☐ | For Bayesian analysis, information on the choice of priors and Markov chain Monte Carlo settings |
| ☒ | ☐ | For hierarchical and complex designs, identification of the appropriate level for tests and full reporting of outcomes |
| ☒ | ☐ | Estimates of effect sizes (e.g. Cohen's *d*, Pearson's *r*), indicating how they were calculated |

*Our web collection on statistics for biologists contains articles on many of the points above.*

## Software and code

Policy information about availability of computer code

| | |
|---|---|
| Data collection | SerialEM v3.8, SerialEM v4.0, Warp v1.0.9 |
| Data analysis | cryoSPARC v3.4.0, FREALIGN v9.11, RELION v3.1, RELION v4.0.1, deepEMhancer (no version number), Chimera v1.15, ChimeraX v1.4, 1.5, or 1.6, IMOD v4.12.37, PEET v1.17.0, SWISS-MODEL (no version number), AlphaFold2, Coot v0.9.8, Phenix v1.19 or 1.20, Namdinator (no version number), ModelAngelo v1.0, findMySequence v0.8.6, DALI server (no version number), DeepTracerID (no version number), FoldSeek (no version number), MOLREP v11, SITUS v3.1, Mascot v2.7.0, Sequest v28.13, custom scripts available from https://github.com/rui--zhang/Doublet |

For manuscripts utilizing custom algorithms or software that are central to the research but not yet described in published literature, software must be made available to editors and reviewers. We strongly encourage code deposition in a community repository (e.g. GitHub). See the Nature Portfolio guidelines for submitting code & software for further information.

## Data

Policy information about availability of data

All manuscripts must include a data availability statement. This statement should provide the following information, where applicable:

- Accession codes, unique identifiers, or web links for publicly available datasets
- A description of any restrictions on data availability
- For clinical datasets or third party data, please ensure that the statement adheres to our policy

Composite cryo-EM maps of the 96-nm repeat of axonemal DMTs from bovine sperm, bovine oviductal cilia, human oviductal cilia, and porcine brain ventricle cilia have been deposited to the EMDB with codes EMD-50664, EMD-45783, EMD-45785, and EMD-45784 respectively. Local refinements for bovine sperm have been deposited to the EMDB with codes EMD-50866 to -50886; for bovine oviduct with codes EMD-45683 to -45697; for human oviduct with codes EMD-45714 to -45725 and -45790; and for porcine brain ventricle with codes EMD-45699 to -45713. Subtomogram averages of the 96-nm repeat of axonemal DMTs from porcine oviductal cilia have been deposited with codes EMD-45677 to -45680. Cryo-EM maps of the 48-nm DMT repeat from bovine oviductal cilia and porcine brain ventricle cilia have been deposited with codes EMD-45801 and -45802. The atomic model of the 96-nm repeat of the bovine sperm DMT is has been deposited to the PDB with accession code 9FQR. Atomic models of the 48-nm DMT repeat from bovine oviductal cilia and porcine brain ventricle cilia have been deposited to the PDB with accession codes 9CPB and 9CPC respectively. Previously-reported atomic models of the 96-nm repeat from human respiratory cilia and of the 48-nm repeat from bovine sperm were used as initial models and are available with PDB accession codes 8J07 and 8OTZ respectively. Proteomics data from bovine oviductal cilia, human oviductal cilia, and porcine brain ventricle cilia are available in Supplementary Table 4. Custom scripts used in this study are publicly available at https://github.com/rui--zhang/Doublet.

## Research involving human participants, their data, or biological material

Policy information about studies with human participants or human data. See also policy information about sex, gender (identity/presentation), and sexual orientation and race, ethnicity and racism.

| | |
|---|---|
| Reporting on sex and gender | Oviduct samples were provided by female organ donors. |
| Reporting on race, ethnicity, or other socially relevant groupings | Data on race, ethnicity and social groupings was not shared with us during procurement or processing of the tissue. |
| Population characteristics | Age, genotypic information and medical history were not shared with us during procurement or processing of the tissue. |
| Recruitment | Participants were registered organ donors. |
| Ethics oversight | The protocol of procurement and processing of human uteri was reviewed by an Institutional Review Board of Harvard University (Protocol# IRB21-0272), which determined the tissue procurement and processing was not human subject research. No identifying information of the deceased organ donors was shared in procurement or processing of the tissue. |

Note that full information on the approval of the study protocol must also be provided in the manuscript.

# Field-specific reporting

Please select the one below that is the best fit for your research. If you are not sure, read the appropriate sections before making your selection.

☒ Life sciences      ☐ Behavioural & social sciences      ☐ Ecological, evolutionary & environmental sciences

For a reference copy of the document with all sections, see nature.com/documents/nr-reporting-summary-flat.pdf

# Life sciences study design

All studies must disclose on these points even when the disclosure is negative.

| | |
|---|---|
| Sample size | For cryo-EM/ET processing, no methods were used to predetermine sample size. The size of the cryo-EM datasets was determined by the need to obtain structures with sufficient resolution to identify differences, and for the bovine sperm doublet microtubule, to build an atomic model. The number of micrographs and particles are listed in the Extended Data. |
| Data exclusions | Micrographs with low resolution estimates following CTF fitting were discarded. Some tilt images were discarded from tomogram reconstruction. The algorithms used for image processing may down-weigh or exclude particles as part of their refinement strategy. |
| Replication | Replication is not necessary for structural studies. Cryo-EM maps represent an average of many thousands of individual copies of the complex of interest, collected from multiple preparations across several microscope sessions. Structures of oviductal cilia doublet microtubules were obtained from three different organisms (bovine, porcine, and human). |
| Randomization | For calculation of the Fourier Shell Correlation (FSC), cryo-EM particles or pseudosubtomograms were randomly split into two halves. |
| Blinding | Blinding is not necessary since there are no groups that need subjective analysis. |

# Reporting for specific materials, systems and methods

We require information from authors about some types of materials, experimental systems and methods used in many studies. Here, indicate whether each material, system or method listed is relevant to your study. If you are not sure if a list item applies to your research, read the appropriate section before selecting a response.

## Materials & experimental systems

| n/a | Involved in the study |
|---|---|
| ☒ | Antibodies |
| ☒ | Eukaryotic cell lines |
| ☒ | Palaeontology and archaeology |
| ☐ | ☒ Animals and other organisms |
| ☒ | Clinical data |
| ☒ | Dual use research of concern |
| ☒ | Plants |

## Methods

| n/a | Involved in the study |
|---|---|
| ☒ | ChIP-seq |
| ☒ | Flow cytometry |
| ☒ | MRI-based neuroimaging |

## Animals and other research organisms

Policy information about studies involving animals; ARRIVE guidelines recommended for reporting animal research, and Sex and Gender in Research

| | |
|---|---|
| Laboratory animals | The study did not involve laboratory animals. |
| Wild animals | The study did not involve wild animals. |
| Reporting on sex | Sperm samples were obtained from male animals. Oviduct samples were obtained from female animals. The sex of the animals of the brain samples is unknown. |
| Field-collected samples | The study did not involve samples collected from the field. |
| Ethics oversight | No ethical approval or guidance was required because organs were used from animals sacrificed for other purposes. Porcine oviductal tissue was acquired from Animal Technologies. Bovine oviducts and porcine brains were sourced from Trenton Processing Center (Trenton, IL) or the Division of Comparative Medicine at Washington University in St. Louis. |

Note that full information on the approval of the study protocol must also be provided in the manuscript.

## Plants

| | |
|---|---|
| Seed stocks | n/a |
| Novel plant genotypes | n/a |
| Authentication | n/a |

