## [Peer Review File · Nature]

Manuscript Title: Structural diversity of axonemes across mammalian motile cilia

Reviewer Comments & Author Rebuttals

Reviewer Reports on the Initial Version:

Referee #1 (Remarks to the Author):

Cilia from various cell types display different waveforms adapted to the different environments they need to function in. A fundamental question in the field has been how the structure of the axoneme is specialized across different cell types. In this manuscript, Leung et al use a combination of cryo-EM, cryo-ET and proteomics to identify the structural differences of axonemal microtubule doublets from sperm flagella, oviductal cilia, brain ventricle ependymal cilia and respiratory cilia. The structures reveal that the axonemal doublets across cilia from various epithelial tissue are very similar and share a common core with the axonemal doublet from the sperm flagella being specialized and containing additional components. This finding sheds light on the evolution of axonemes indicating that specialization arises from the addition of components and modification of a common evolutionarily shared core. Highlights of the manuscript include: 1) a better than 5 Å resolution EM map of the radial spoke 3 (RS3) in sperm flagella that allowed model building and revealed the regular arrangement of several kinases involved in ATP regeneration, required to maintain the high concentrations of ATP needed during axonemal beating; 2) the identification of the TRiC chaperon complex ((7.4 Å) tethered to RS1 and RS2 in the sperm axonemal structure, but absent in the doublet structures from epithelial cilia; 3) resolving the inner dynein arm (IDA) in the pre-powerstroke state (at 4-6 Å) and shedding light on the conformational changes between the pre- and post-powerstroke conformations. This work is important and will be of interest to cilia biologists, scientists interested in the cytoskeleton and scientist interested the structure and assembly of large macromolecular complexes. The quality of the work is high and the manuscript is overall well written. Therefore, I am very supportive of publication; however, there are several important aspects that the authors need to address.

1. The authors provide workflows of EM/ET data processing in Extended Data Figure 1, but it is not clear how the local classification and refinement were done. As the authors noted in the manuscript, sperm axonemes display structural asymmetry, especially in the RS region (Chen et al, NSMB 2023). Some DMTs have extra densities, including TRiC, while others don't. Did the authors examine which DMTs they are reporting here if comparing their composite maps to the subtomogram averages from the Chen 2023 study? Or the composite map is a mixture of features from all DMTs and the information could not be recovered during data processing? If this is the case, the authors should note it in the manuscript. The authors should also provide a scheme of how the local classification and refinement was done in a supplementary figure, ie what classes they obtained and how many rounds of subtraction, classification and refinement they did. The same for axonemal dynein – please provide the information on the percentage of particles in the pre-powerstroke vs post-powerstroke conformation in Methods or SI.

2. It is not clear from the Methods how the cryo-ET data was used in the processing. In general, the Methods are cursory. It is important that Methods are detailed enough for others to be able to reproduce the work.

3. Define what the 96 versus 48 nm repeat is for the non-specialist reader.

4. The authors should provide a complete gallery of the 34 newly identified proteins with examples of map quality and model fit, especially in RS3. This can be provided in a supplementary figure.

5. The authors compare their TRiC chaperone complex with the yeast TRiC structure and discovered that the luminal density they find is not in the same place (Han et al 2023). Have they looked at other TRiC structures which show substrate tubulin or actin bound? For example Gestuat et al 2022 or Kelly et al 2022.

Can the authors tell whether their TRiC is in an open or closed conformation? Or perhaps it is a mixture of open/closed states that limits the local resolution? The resolution of the TriC chaperone is only at $\sim 7\text{\AA}$. Have the authors tried to classify the TriC particles to see whether they can separate different conformational states?

Also, the section that presents the TriC structure ends abruptly. Can the authors speculate as to why the TriC is associated in this way, or why it is regularly spaced in sperm cilia and not epithelial cilia?

6. They find PKA organized as a dimer consisting of the regulatory and catalytic subunit, not a tetrameric arrangement like in crystal structures. Are there functional consequences to this arrangement?

Is the binding affinity between the catalytic subunit of the PKA and the regulatory subunit compatible with a model of it dissociating and diffusing in the cilia as they propose? A sentence or two of clarification here based on what is known about this regulatory complex from other systems would be helpful.

7. It might help the reader if the ciliary substrates of the kinases identified in this study (PKA, CAMK4 and STKLD1) can be summarized in a supplementary table.

8. No data has been deposited to the PDB and EMDB. The deposition codes need to be supplied with the revised manuscript. Also, it is unclear which datasets were collected for this work and what datasets were collected previously.

9. Custom scripts which they used in processing need to be provided or uploaded to a public repository.

10. The fit of the nucleotide in 4b is not visible. Please provide a larger figure in SI that shows the fit of the nucleotide clearly.

11. In the extended data figure 7b legend, the authors state "... potentially induced by binding RGS22." (page 23). Do the authors mean EFCAB5?

Referee #2 (Remarks to the Author):

In this study, the authors extend on their elegant structural studies of axonemes from different species. Novelty in this work comes from a more complete single-particle cryo-EM map and atomic model of the bovine sperm doublet microtubule. Previous investigations from the authors (PMID: 37327785) and others (e.g. PMIDs: 37295417, 37865089, 37989994) focussed on microtubule inner proteins (MIPs) and microtubule associated proteins (MAPs) near the surface of the sperm doublet microtubule. Here, more peripheral structures are resolved, enabling an atomic model of the third radial spoke (RS3) and a putative TRiC chaperone linking RS1 and RS2, among other proteins. The authors also find two different conformations of dynein-f in their bovine sperm doublet microtubule sample, one in which dynein-f contacts the adjacent nexin-dynein regulatory complex. Atomic modelling focusses on the bovine sperm doublet microtubule, but single-particle cryo-EM maps are also presented for doublet microtubules from ependymal cilia (porcine) and oviduct cilia (human and bovine), together with a cryo-ET map of oviduct cilia (porcine). Comparison is made with the authors' previous respiratory cilia maps (bovine and human), leading to the conclusion that the sperm structure is the outlier whereas the others are broadly similar. This is a large body of work, laying further valuable foundations for understanding the molecular mechanism of axonemal beating in different cilia types, which remains poorly understood and of high physiological importance.

Main comments:

- Although the authors have gone to commendable effort to give the local resolution ranges for the different complexes (Table S2) and rationale for the protein assignments (Table S5), it remains difficult for the reader to judge the confidence level in each protein assignment. A model-map FSC, CC score, or equivalent metric would help judge how well each atomic model fits the map. The model-in-map images shown in the Extended Data Figures are too small to tell if the protein assignments are unambiguous. In cases in which initial backbone traces were compared to databases, it also would be valuable to know how unique the top hit was (i.e. how much better did it score compared to other hits, and, if there was little to separate them, what additional evidence was used to discard other hits).

- In the case of barrel-shaped density, the assignment of CCT1-8 does not look clear cut from what the reader is shown. The atomic model predicts contacts between CCT3 and AK8, and CCT6 and RGS22. Can this be tested by AlphaFold Multimer prediction of these interfaces? Does AlphaFold Multimer shed any light on the incorporation of the sperm-specific TRiC paralogs, CCT6B and CCT8L2?

- The different conformations of ODA and IDAf are among the most intriguing observations, and raise several questions.

-- The bovine sperm sample (and previous samples) are in 1 mM ATP, so it is somewhat surprising that the majority of the dyneins have straight linkers. In the case of cytoplasmic dynein, the release of hydrolysis products is rate limiting, causing dyneins to accumulate in the bent linker conformation in the presence of ATP (e.g. PMID: 15880123). Although it may be challenging at the 4.6 Å angstrom resolution of the IDAf maps, can the authors say anything about the nucleotide state of the motors, and what may be causing their different conformations? It would also be helpful to know what

fraction of particles are in each of the two conformations, and if the conformations in one 96 nm repeat are correlated with those nearby. Biologically, it is important to start correlating the conformational changes with nucleotide state, especially given the observation that in actively beating cilia most dyneins appear to have bent linkers (PMID: 29700238).

-- In the case of IDAf, naming the conformations pre- and post-powerstroke is problematic, given that the powerstroke is the force generating conformational change that the motor undergoes while bound to its track (i.e. the true pre-powerstroke conformation is when the motor has re-attached to its track but not yet undergone the conformational change). In the case of IDAf, it is not yet clear if it functions as a motor and undergoes a powerstroke on the microtubule, based on its activity in vitro (especially for the alpha chain; PMID 17496036 and 21148301). This should be cited. Even if IDAf does undergo a powerstroke, in the state captured in this study, it is not bound to the B-microtubule, so cannot be the true pre-powerstroke state. Referring to it as pre-state, primed state, or bent linker state would most appropriate. A small but important change for accuracy.

-- The authors capture a state in which the microtubule-binding domain of the IDAf alpha contacts the N-DRC, further challenging the idea that this dynein functions as a microtubule motor. It would be interesting to know if this contact is supported by an AlphaFold Multimer prediction of the interface. It would also be valuable to hear the authors thoughts on whether the microtubule-binding domains of either IDAf alpha or beta could reach the B-microtubule in the primed state, and whether their structure is more consistent with IDAf acting as a communication device or a microtubule motor.

-- While most figures do a good job of guiding the reader through the complexity, this is not currently the case in Figure 6. The conformational changes of IDAf are somewhat lost amongst the other brightly colored densities, and the switches between ribbon representation, surface representation, and EM densities makes it difficult to relate the panels to each other and are not explained in the legend.

Other comments:

Cryo-ET map

- Is there evidence for asymmetries between the doublets in the cryo-ET maps from porcine oviduct cilia?

"The divergent C-terminus of CFAP91 found in Chlamydomonas likely explains the absence of RS3 in green algae"

- It does not appear possible to assign causality here i.e. RS3 could have been lost in green algae by another mechanism, followed by divergence of the CFAP91 C terminus.

"Binding to EFCAB5 appears to induce a conformational change in RGS22, which adopts a more extended conformation in RS2 than in RS1 (Extended Data Fig. 7b)"

- Colors in Extended Data Fig. 7b are too close to differentiate.

Anchored kinases

- As PKA^C is tethered to RS3 solely by PKA^R, the authors propose that the catalytic subunit is released upon cAMP binding. In the case of the other RS-anchored kinases, CAMK4 and STKLD1, do the authors envision that these are also released, or are they too buried, meaning that substrates would have to diffuse to them within the RS?

Extended Data Figure 8b

- Do the unfilled extra densities derive from a difference map? If so, this should be clarified. A negative control difference map (i.e. using a nearby protein not expected to bind small molecules, which should show no extra densities) could strengthen the argument that the extra densities in the open state are real.

"the motor and stalk of f β interact with the linkers of IDAd and IDAg respectively (Fig. 6d)"

- Label linkers of IDAd and IDAg in Fig 6d

- Label TEX47 in in Fig 6b

Referee #3 (Remarks to the Author):

This structural biology work uses cryo-EM to reveal molecular organization of motile cilia, beating organelle, from mammals. Recently this field experienced great progress, enabling unprecedented atomic or pseudo-atomic resolution 3D imaging of >100 proteins by single particle cryo-EM. The approach is to analyze the 96nm microtubule-based periodic unit as a single particle (Walton et al. (2023) Nature 618, 625), which revealed atomic structures of ciliary doublet microtubules from green algae and from human, providing enormous encyclopedic information of ciliary structure. The same cryo-EM approach was employed by the following works, addressing ciliary doublet from sperm (Zhou et al. (2023) Cell 186, 2897; Leung et al. (2023) Cell 186, 2880). These works shed light on the special protein composition, unique for sperm compared to other cilia (which was studied in Gui et al. (2021) Cell 184, 5791), and correlated the structural information to functional analysis of newly characterized mutants (ciliopathy patients) caused by defect of these proteins. Enormous success of single particle cryo-EM work enabled these structure-function correlation.

This work is heading the same direction, addressing structural characterization of mammalian ciliary doublet structure from sperm and epithelial cells. This work contains several novel findings based on high resolution structures of mammalian ciliary doublet microtubules from various organs and various species. The major results of this work are (1) comparison of ciliary structures with conclusion that sperm cilia are most different with 34 unique proteins, while epithelial cilia are similar to each other, (2) atomic resolution reconstruction of the novel 3D structure of RS3 (the third radial spoke), which was not known, and identification of AKs and PKA, (3) identification of TRiC chaperone at the radial spoke, which was not identified due to lack of resolution in the past, and (4) alternative structure of dynein f, which is likely pre-powerstroke conformation at atomic resolution. In the viewpoint of this reviewer, (1) provides additional protein composition information, after precise analysis by Zhou et al. and Leung et al. revealing most of the proteins. This additional protein information will be attractive for cilia researchers. (2) and (3) are exciting findings of enzymes, which fit to the functions we expected but could not be located until now, and lead us to nice hypothetical stories how ATP is regenerated and how chaperone works at cilia. (4) present higher resolution view of dynein f conformational change than the past work by Lin and Nicastro (2018) Science 360, 1. Since this work, despite of beautiful illustrations and abundant information, is limited to structural studies of missing pieces with speculative stories solely based on the structure, this reviewer thinks it is suitable for journals specific for molecular structural biology, such as Nature Structural and Molecular Biology (as the authors published in the past NSMB (2022) 29, 483), Molecular Cell or EMBO Journal. With additional studies to prove functions of TRiC, AK or PKA, this work could be attractive for wide readers of Nature.

Nevertheless this reviewer would advise the authors for possible revision to improve the manuscript.

Major points:

Many newly found proteins are only briefly mentioned. For example, in lines 131 and 132. The authors should be able to describe their conformation, interaction with neighboring proteins and possible functions.

Line132-135, Fig.2b: relationship between the data in this study and ciliopathy is not clear. This reviewer could not find the correlation between sperm specific proteins and sperm specific diseases. More explanation needed.

Supplementary Table 2 and Line137-: RS3 molecular modeling

This reviewer is not fully convinced that protein identification can be done at >4Å resolution. In Suppl. Table 2, mostly structure and abundance of protein in cilia are used as the basis. Can the author provide density maps used for chain tracing in EMDB? Can they compare mammalian cilia and Chlamydomonas by proteomics and show absence of RS3 proteins in Chlamydomonas, which lacks RS3?

Line184-190: Do these molecules have homologues in Chlamydomonas? If not, do mammalian cilia need more ATP than protists?

Line196-200: open and closed states of AKs

How do the authors think two conformations co-exist? Do they show two stages of catalytic cycle? Or just fluctuation? If the authors think it is catalytic, can they shift equilibrium between two states biochemically (by adding phosphate?) and prove it by cryo-EM and classification?

Line329-330 and Supplementary Table 8: It is not clear how this table presents earlier or later expression. More explanation is needed.

Minor points:

Line84: Remarkable consistency with our SPA structure

Zimmermann et al. (2023) EMBO J. 42, e112466 claims ODA structure by cryo-ET needed remodeling from SPA structures. Did the author find no such slight difference between SPA from sprayed axoneme and cryo-ET of intact axoneme?

Line126: new proteins across nearly every axonemal complex, including an ARMH1 subcomplex that is distributed asymmetrically around the axoneme

Did author find ARMH1 for every doublet, while cryo-ET in the past work showed asymmetrical arrangement? What is the reason of the past work missing ARMH1? Did they conduct classification to map the 96nm repeat with and without ARMH1?

Line263: How was pre-powerstroke conformation made? Did the author mix ATP in the specimen? Or was it activated by endogenous ATPs?

Author Rebuttals to Initial Comments:

We thank the referees for their detailed and helpful feedback. We were thrilled to read their positive appraisal of our work. We have endeavoured to address the referees' suggestions to the best of our ability; below are our responses to their comments. We have also marked changes in blue throughout the manuscript and SI documents.

Point-by-point response

Referee #1

Cilia from various cell types display different waveforms adapted to the different environments they need to function in. A fundamental question in the field has been how the structure of the axoneme is specialized across different cell types. In this manuscript, Leung et al use a combination of cryo-EM, cryo-ET and proteomics to identify the structural differences of axonemal microtubule doublets from sperm flagella, oviductal cilia, brain ventricle ependymal cilia and respiratory cilia. The structures reveal that the axonemal doublets across cilia from various epithelial tissue are very similar and share a common core with the axonemal doublet from the sperm flagella being specialized and containing additional components. This finding sheds light on the evolution of axonemes indicating that specialization arises from the addition of components and modification of a common evolutionarily shared core. Highlights of the manuscript include: 1) a better than 5 Å resolution EM map of the radial spoke 3 (RS3) in sperm flagella that allowed model building and revealed the regular arrangement of several kinases involved in ATP regeneration, required to maintain the high concentrations of ATP needed during axonemal beating; 2) the identification of the TRiC chaperon complex ((7.4 Å) tethered to RS1 and RS2 in the sperm axonemal structure, but absent in the doublet structures from epithelial cilia; 3) resolving the inner dynein arm (IDA) in the pre-powerstroke state (at 4-6 Å) and shedding light on the conformational changes between the pre- and post-powerstroke conformations. This work is important and will be of interest to cilia biologists, scientists interested in the cytoskeleton and scientist interested the structure and assembly of large macromolecular complexes. The quality of the work is high and the manuscript is overall well written. Therefore, I am very supportive of publication; however, there are several important aspects that the authors need to address.

We thank the reviewer for their positive assessment of our work.

1. The authors provide workflows of EM/ET data processing in Extended Data Figure 1, but it is not clear how the local classification and refinement were done. As the authors noted in the manuscript, sperm axonemes display structural asymmetry, especially in the RS region (Chen et al, NSMB 2023). Some DMTs have extra densities, including TRiC, while others don't. Did the authors examine which DMTs they are reporting here if comparing their composite maps to the subtomogram averages from the Chen 2023 study? Or the composite map is a mixture of features from all DMTs and the information could not be recovered during data processing? If this is the case, the authors should note it in the manuscript.

All composite maps reported in our study, from both SPA cryo-EM and cryo-ET, represent averages of all nine DMTs. We now explicitly state this in the Main Text and Methods.

Unfortunately, we cannot confidently ascertain whether specific classes resolved by our SPA approach correspond to specific DMTs for two main reasons: (1) the asymmetric arrangement of axonemal complexes differs across species (Chen et al 2023 PMID 36593309), so it may not be possible to deduce DMT identity in bovine sperm by direct comparison to mouse or human sperm; (2) absence of structures in certain classes may also be due to protein loss during sample preparation for SPA. Nonetheless, by providing the molecular identities of some of the asymmetrically distributed components, our work should help facilitate future studies of axonemal asymmetry.

The authors should also provide a scheme of how the local classification and refinement was done in a supplementary figure, ie what classes they obtained and how many rounds of subtraction, classification and refinement they did. The same for axonemal dynein – please provide the information on the percentage of particles in the pre-powerstroke vs post-powerstroke conformation in Methods or SI.

We apologize for the missing processing details in the initial submission. We have now expanded the Methods and added new Supplementary Figures 1-7 that show processing schemes for local regions of the bovine sperm DMT. In each figure, we show the starting reconstruction, the mask applied, the results of three-dimensional classification including particle distributions, the resolution following refinement (according to the FSC=0.143 criterion), and the corresponding FSC curves. As illustrated in Supplementary Figure 6, ~31% of the particles are in the pre-powerstroke state. We couldn't confidently determine if the remaining 69% particles are in the post-stroke state or other intermediate state.

2. It is not clear from the Methods how the cryo-ET data was used in the processing. In general, the Methods are cursory. It is important that Methods are detailed enough for others to be able to reproduce the work.

Our apologies that this was unclear. Cryo-ET data was used for subtomogram averaging to generate a reconstruction of the 96-nm repeat from porcine oviduct cilia; all other structures were calculated using single-particle data alone. We have added text clarifying this to the “Cryo-ET data processing” section of the Methods.

We have generated seven new Supplementary Figures and expanded the Methods text to include more details of our processing strategies.

3. Define what the 96 versus 48 nm repeat is for the non-specialist reader.

We have added a sentence to the Introduction to clarify the difference between the 96-nm and 48-nm repeats:

“In addition, the DMT lumen is extensively decorated with microtubule inner proteins (MIPs) that bind in varying multiples of the 8-nm tubulin repeat, but with an overall periodicity of 48-nm that is in coherent register with the external 96-nm repeat.”

4. The authors should provide a complete gallery of the 34 newly identified proteins with examples of map quality and model fit, especially in RS3. This can be provided in a supplementary figure.

We have generated a supplementary document (Data S1) that contains a separate page for each newly identified protein. Each page includes the database ID of the sequence used for modeling, the location of the protein in the 96-nm repeat, its abundance (iBAQ) in the bovine sperm mass spectrometry data, a model-in-map figure, a correlation coefficient, and the rationale for protein assignment (including identification strategies and supporting evidence from the literature).

5. The authors compare their TRiC chaperone complex with the yeast TRiC structure and discovered that the luminal density they find is not in the same place (Han et al 2023). Have they looked at other TRiC structures which show substrate tubulin or actin bound? For example Gestuat et al 2022 or Kelly et al 2022.

We have compared our structure with several others (Rebuttal Figure 1). This shows that the luminal density in RS-tethered TRiC binds to a location distinct from substrate or from co-chaperones such as PhLP1, plp2, or prefoldin.

Rebuttal Figure 1. Comparison of luminal density in RS-tethered TRiC (a) with densities for TRiC substrates and co-chaperones. Structures compared are: (b) yeast TRiC in complex with the co-chaperone plp2 (EMD-33918, Han et al 2023 PMID 36921056); (c) human TRiC in complex with substrate (G β 5) and the co-chaperone PhLP1 (EMD-40439, Wang et al 2023 PMID 37852256); (d) human TRiC in complex with substrate (actin/tubulin) and a nanobody (EMD-12608, Kelly et al 2022 PMID 35449234); (e) human TRiC in complex with substrate (β -tubulin) and prefoldin (EMD-32823, Gestaut et al 2022 PMID 36493755).

Can the authors tell whether their TRiC is in an open or closed conformation? Or perhaps it is a mixture of open/closed states that limits the local resolution? The resolution of the TriC chaperone is only at $\sim 7\text{\AA}$. Have the authors tried to classify the TriC particles to see whether they can separate different conformational states?

The RS-tethered TRiC appears to be in the open conformation (Rebuttal Figure 2). 3D-classification did not resolve any classes resembling the closed state.

Rebuttal Figure 2. Comparison with published TRiC structures suggests that the RS-tethered TRiC is in the open state. The map of open-state TRiC is EMD-0492 (Gestaut et al 2019 PMID 309558883) and the map of closed-state TRiC is EMD-26089 (Gestaut et al 2022 PMID 36493755).

Also, the section that presents the TriC structure ends abruptly. Can the authors speculate as to why the TriC is associated in this way, or why it is regularly spaced in sperm cilia and not epithelial cilia?

One plausible explanation is that the protein tethering TRiC to RS2 (i.e. the grey “linker” in Figure 5c) is only expressed in sperm, although confirmation awaits definitive identification of the linker. We have stated this possibility in the text.

6. They find PKA organized as a dimer consisting of the regulatory and catalytic subunit, not a tetrameric arrangement like in crystal structures. Are there functional consequences to this arrangement? Is the binding affinity between the catalytic subunit of the PKA and the regulatory subunit compatible with a model of it dissociating and diffusing in the cilia as they propose? A sentence or two of clarification here based on what is known about this regulatory complex from other systems would be helpful.

In purified PKA, the docking/dimerization (D/D) domains of the regulatory subunits homodimerize, which leads to the tetrameric arrangement observed in the crystal structure. In contrast, axoneme-anchored PKA is a dimer likely because the D/D domain of the regulatory subunit interacts with a D/D domain of another RS3 protein. While we cannot definitively assign the D/D domain of the PKA regulatory subunit at the resolution of our maps, one intriguing possibility is that it dimerizes with the C-terminal D/D domain of CATIP, which pairs with an unassigned density.

A model in which PKA^C diffuses from PKA^R during signaling is consistent with evidence from neurons (Tillo et al 2017 PMID 28423323, Xiong et al 2024 PMID 37732264) and immortalized cell lines (Walker-Gray et al 2017 PMID 28893983), along with other biochemical and *in vitro* assays (reviewed in Gold 2019 PMID 31671183). Diffusion of the catalytic subunit would also be consistent with super-resolution microscopy data in mouse sperm showing that PKA^C distribution changes from a tight cylinder of mean radius ~70 nm in non-capacitated sperm to a broader ~105 nm radius in capacitated sperm (Stival et al 2018 PMID 29700114). We

have expanded the text under the section “Radial spoke-anchored signaling protein kinases” to reference these studies.

7. It might help the reader if the ciliary substrates of the kinases identified in this study (PKA, CAMK4 and STKLD1) can be summarized in a supplementary table.

To our knowledge, the axonemal substrates of PKA, CAMK4, and STKLD1 have not been precisely defined experimentally. To do so would require careful phosphoproteomic analysis of sperm treated with specific kinase inhibitors under capacitating and non-capacitating conditions, which would warrant a separate comprehensive study.

8. No data has been deposited to the PDB and EMDB. The deposition codes need to be supplied with the revised manuscript. Also, it is unclear which datasets were collected for this work and what datasets were collected previously.

We have deposited models to the PDB and the composite and locally refined maps to the EMDB. Accession codes are provided in the “Data availability” section and in Supplementary Tables 1 and 2.

The datasets of oviductal and brain ventricle cilia were newly collected for this study. For bovine sperm, we collected 34,796 new movies in addition to the 10,635 movies reported previously (Leung & Zeng et al 2023 PMID: 37327785). We have clarified this in the Methods.

9. Custom scripts which they used in processing need to be provided or uploaded to a public repository.

Custom scripts used in this study are publicly available at <https://github.com/rui--zhang/Doublet>. We have added a “Code availability” statement to the manuscript to reflect this.

10. The fit of the nucleotide in 4b is not visible. Please provide a larger figure in SI that shows the fit of the nucleotide clearly.

We have added a larger panel to the Data S1 page of CAMK4 that shows the fit of the putative nucleotide more clearly.

11. In the extended data figure 7b legend, the authors state “... potentially induced by binding RGS22.” (page 23). Do the authors mean EFCAB5?

Yes, we meant EFCAB5 – we have corrected this in the figure legend.

Referee #2

In this study, the authors extend on their elegant structural studies of axonemes from different species. Novelty in this work comes from a more complete single-particle cryo-EM map and atomic model of the bovine sperm doublet microtubule. Previous investigations from the authors (PMID: 37327785) and others (e.g. PMIDs: 37295417, 37865089, 37989994) focussed on microtubule inner proteins (MIPs) and microtubule associated proteins (MAPs) near the surface of the sperm doublet microtubule. Here, more peripheral structures are resolved, enabling an atomic model of the third radial spoke (RS3) and a putative TRiC chaperone linking RS1 and RS2, among other proteins. The authors also find two different conformations of dynein-f in their bovine sperm doublet microtubule sample, one in which dynein-f contacts the adjacent nexin-dynein regulatory complex. Atomic modelling focusses on the bovine sperm doublet microtubule, but single-particle cryo-EM maps are also presented for doublet microtubules from ependymal cilia (porcine) and oviduct cilia (human and bovine), together with a cryo-ET map of oviduct cilia (porcine). Comparison is made with the authors' previous respiratory cilia maps (bovine and human), leading to the conclusion that the sperm structure is the outlier whereas the others are broadly similar. This is a large body of work, laying further valuable foundations for understanding the molecular mechanism of axonemal beating in different cilia types, which remains poorly understood and of high physiological importance.

We thank the reviewer for their positive assessment of our work.

Main comments:

- Although the authors have gone to commendable effort to give the local resolution ranges for the different complexes (Table S2) and rationale for the protein assignments (Table S5), it remains difficult for the reader to judge the confidence level in each protein assignment. A model-map FSC, CC score, or equivalent metric would help judge how well each atomic model fits the map. The model-in-map images shown in the Extended Data Figures are too small to tell if the protein assignments are unambiguous. In cases in which initial backbone traces were compared to databases, it also would be valuable to know how unique the top hit was (i.e. how much better did it score compared to other hits, and, if there was little to separate them, what additional evidence was used to discard other hits).

We have generated a supplementary document (Data S1) that contains a separate page for each newly identified protein. Each page includes the database ID of the sequence used for modeling, the location of the protein in the 96-nm repeat, its abundance (iBAQ) in the bovine sperm mass spectrometry data, a model-in-map figure, a correlation coefficient, and the rationale for protein assignment (including identification strategies and supporting evidence from the literature). This document replaces Table S5.

- In the case of barrel-shaped density, the assignment of CCT1-8 does not look clear cut from what the reader is shown. The atomic model predicts contacts between CCT3 and AK8, and CCT6 and RGS22. Can this be tested by AlphaFold Multimer prediction of these interfaces? Does AlphaFold Multimer shed any light on the incorporation of the sperm-specific TRiC paralogs, CCT6B and CCT8L2?

We assigned subunit order based on certain features that are consistent across several open-state TRiC structures (Liu et al 2023 PMID 37193829), specifically that CCT1 tends to be tilted outwards and that the largest gap in the ring is found between CCT1 and CCT4. Since subunit order is highly conserved across known TRiC structures, assigning CCT1 and CCT4 allowed us to assign the neighboring densities to the remaining CCT subunits.

To confirm our assignment, we took a structure of open-state TRiC (PDB 7YLV, Han et al 2023 PMID 36921056), fitted the eight possible rotations into our cryo-EM maps, and calculated the resulting average

map values and CC scores in ChimeraX (Rebuttal Figure 3). The highest score matched our initial assignment. We have added this analysis to the Data S1 page for TRiC.

Rebuttal Figure 3. Assessing possible fits of open-state TRiC (PDB 7YLV) into the cryo-EM density.

AlphaFold predictions for TRiC paralogs are very similar, consistent with the high sequence identity between them (87% for CCT6/6B and 31% for CCT8/8L2) (Rebuttal Figure 4a-b). AlphaFold3 predictions also suggest that CCT6B and CCT8L2 would occupy the same sites as CCT6A and CCT8 in the ring (Rebuttal Figure 4c). We also tried using AlphaFold3 to predict the interfaces between CCT3/AK8 and between CCT6/RGS22, but these attempts did not yield robust complexes. This may be because the interfaces are relatively small and/or require the initial positioning and pre-orientation of TRiC by tethering to RS2 via an as-yet-unidentified linker.

a CCT6A CCT6B

b CCT8 CCT8L2

c

Rebuttal Figure 4. (a-b) AlphaFold predictions and sequence alignments for (a) CCT6 and CCT6B and (b) CCT8 and CCT8L2. (c) AlphaFold3 predictions suggest that CCT6B and CCT8L2 (right panel) would occupy the same sites as CCT6A and CCT8 (left panel) respectively.

- The different conformations of ODA and IDAf are among the most intriguing observations, and raise several questions.

-- The bovine sperm sample (and previous samples) are in 1 mM ATP, so it is somewhat surprising that the majority of the dyneins have straight linkers. In the case of cytoplasmic dynein, the release of hydrolysis products is rate limiting, causing dyneins to accumulate in the bent linker conformation in the presence of ATP (e.g. PMID: 15880123). Although it may be challenging at the 4.6 Å angstrom resolution of the IDAf maps, can the authors say anything about the nucleotide state of the motors, and what may be causing their different conformations? It would also be helpful to know what fraction of particles are in each of the two conformations, and if the conformations in one 96 nm repeat are correlated with those nearby. Biologically, it is important to start correlating the conformational changes with nucleotide state, especially given the observation that in actively beating cilia most dyneins appear to have bent linkers (PMID: 29700238).

We carefully inspected our maps, and while we see density consistent with nucleotides, we cannot confidently assign nucleotide state (Rebuttal Figure 5). Thus, we have not included nucleotides in the deposited model and cannot make statements about the nucleotide state of the pre-state at this point.

Rebuttal Figure 5. Densities in the nucleotide binding pockets of the AAA domains in DNAH10 (*IDAf α*) and DNAH2 (*IDAf β*). For illustration, ATP is shown fitted into AAA2 and ADP in AAA1, AAA3, and AAA4, but we cannot confidently assign nucleotide identity with the current density.

We have summarized the processing scheme for the pre-state in a new Supplementary Figure 6. Approximately 31% of particles showed clear density for the *IDAf β* motor near the base of RS3.

-- In the case of *IDAf*, naming the conformations pre- and post-powerstroke is problematic, given that the powerstroke is the force generating conformational change that the motor undergoes while bound to its track (i.e. the true pre-powerstroke conformation is when the motor has re-attached to its track but not yet undergone the conformational change). In the case of *IDAf*, it is not yet clear if it functions as a motor and undergoes a powerstroke on the microtubule, based on its activity *in vitro* (especially for the alpha chain; PMID 17496036 and 21148301). This should be cited. Even if *IDAf* does undergo a powerstroke, in the state captured in this study, it is not bound to the B-microtubule, so cannot be the true pre-powerstroke state. Referring to it as pre-state, primed state, or bent linker state would most appropriate. A small but important change for accuracy.

We have replaced “pre-powerstroke” and “post-powerstroke” with “pre-state” and “post-state” respectively whenever we refer to structures derived from single-particle cryo-EM. We have also changed corresponding labels in the figures and videos.

We have now cited Kotani et al 2007 PMID 17496036 and Toba et al 2011 PMID 21148301 at the end of the section “Dynein pre-state conformations”. We have also added emphasis that its major role in the axoneme may be as a mechanochemical regulator.

-- The authors capture a state in which the microtubule-binding domain of the IDA α contacts the N-DRC, further challenging the idea that this dynein functions as a microtubule motor. It would be interesting to know if this contact is supported by an AlphaFold Multimer prediction of the interface. It would also be valuable to hear the authors thoughts on whether the microtubule-binding domains of either IDA α or beta could reach the B-microtubule in the primed state, and whether their structure is more consistent with IDA α acting as a communication device or a microtubule motor.

We tested this, but AlphaFold does not robustly predict the interaction between DRC9 and the microtubule-binding domain of DNAH10 (IDA α), likely because it is a transient interaction that only forms under specific conditions (e.g. with the right conformation of the DNAH10 MTBD and supported by the physical proximity of the two complexes in the axoneme).

The microtubule-binding domains (MTBDs) of pre-state IDA α , as observed in our study, would not be capable of reaching the B tubule of the neighboring DMT (Rebuttal Figure 6). In the post-state, the MTBDs should be able to reach the adjacent DMT, which is also consistent with evidence that isolated IDA α can bundle microtubules *in vitro* (Toba et al 2011 PMID 21148301). Note, however, that we could not find a subtomogram average in the EMDB where this interaction is captured convincingly.

Our structures emphasize that IDA α is clearly a major mechanochemical regulator and coordinator of neighboring complexes, including IDA α , IDA β /g, and the N-DRC. However, we cannot exclude that it is also acting as a force-generating motor in some capacity. It is possible that some of the intermediate states reported in Lin & Nicastro 2018 PMID 29700238 represent conformations where IDA α binds the adjacent DMT, but confirmation awaits larger datasets and high-resolution *in situ* cryo-ET studies of actively-cycling IDA α .

Rebuttal Figure 6. Modeling how IDA α interacts with the B-tubule of the neighboring DMT in the post-state (a) and pre-state (b). A subtomogram average from intact *Tetrahymena* axonemes (EMD-9023, Song et al 2019 PMID 31768058) was used to model the position of the neighboring DMT.

-- While most figures do a good job of guiding the reader through the complexity, this is not currently the case in Figure 6. The conformational changes of IDAf are somewhat lost amongst the other brightly colored densities, and the switches between ribbon representation, surface representation, and EM densities makes it difficult to relate the panels to each other and are not explained in the legend.

We have revised Figure 6 based on the referee's suggestions. Specifically, we (i) simplified the coloring in 6a, (ii) added new panels emphasizing the conformational changes in the IDAf motors, and (iii) replaced the EM density representation with surface and ribbon representations for consistency with other panels.

Other comments:

Cryo-ET map

- Is there evidence for asymmetries between the doublets in the cryo-ET maps from porcine oviduct cilia?

Our current subtomogram average represents a consensus of all DMTs, similar to our single-particle cryo-EM maps. We now state this explicitly in the Methods (under the "Cryo-ET data processing" section).

The excellent 2023 paper by Chen and colleagues (PMID: 36593309) sets the benchmark for how structural studies examining the asymmetry of axonemes should be performed. To achieve independent reconstructions of each DMT, the authors needed to first reconstruct a structure of the central apparatus. By expanding the box around this reconstruction, they were then able to re-extract subvolumes corresponding to individual DMTs.

Attempting to perform a similar analysis with our cryo-ET data of porcine oviductal cilia would require completely reprocessing the dataset to first obtain a structure of the central apparatus. We feel that this work is beyond the scope of the current study, which is focused on comparing consensus structures from different motile-ciliated cell types and not asymmetry.

"The divergent C-terminus of CFAP91 found in Chlamydomonas likely explains the absence of RS3 in green algae"

- It does not appear possible to assign causality here i.e. RS3 could have been lost in green algae by another mechanism, followed by divergence of the CFAP91 C terminus.

We have removed this remark and the associated panel.

"Binding to EFCAB5 appears to induce a conformational change in RGS22, which adopts a more extended conformation in RS2 than in RS1 (Extended Data Fig. 7b)"

- Colors in Extended Data Fig. 7b are too close to differentiate.

We have changed the colors in Extended Data Figure 7 to improve contrast.

Anchored kinases

- As PKA^C is tethered to RS3 solely by PKA^R, the authors propose that the catalytic subunit is released upon cAMP binding. In the case of the other RS-anchored kinases, CAMK4 and STKLD1, do the authors envision that these are also released, or are they too buried, meaning that substrates would have to diffuse to them within the RS?

CAMK4 and STKLD1 seem unlikely to diffuse given their extensive interactions with neighboring proteins. Either their targets diffuse to them as the reviewer suggests, or they phosphorylate flexible regions of nearby proteins.

Extended Data Figure 8b

- Do the unfilled extra densities derive from a difference map? If so, this should be clarified. A negative control difference map (i.e. using a nearby protein not expected to bind small molecules, which should show no extra densities) could strengthen the argument that the extra densities in the open state are real.

The extra densities were derived by displaying density within 3-Å of the ligand atoms of a superposed AK crystal structure. We have stated this explicitly in the figure legends for both Extended Data Figure 8 and Main Figure 4.

"the motor and stalk of $f\beta$ interact with the linkers of IDAd and IDAg respectively (Fig. 6d)"

- Label linkers of IDAd and IDAg in Fig 6d

- Label TEX47 in in Fig 6b

We have added additional labels to Figure 6 as requested. We also corrected "linkers" to "tails".

Referee #3

This structural biology work uses cryo-EM to reveal molecular organization of motile cilia, beating organelle, from mammals. Recently this field experienced great progress, enabling unprecedented atomic or pseudo-atomic resolution 3D imaging of >100 proteins by single particle cryo-EM. The approach is to analyze the 96nm microtubule-based periodic unit as a single particle (Walton et al. (2023) Nature 618, 625), which revealed atomic structures of ciliary doublet microtubules from green algae and from human, providing enormous encyclopedic information of ciliary structure. The same cryo-EM approach was employed by the following works, addressing ciliary doublet from sperm (Zhou et al. (2023) Cell 186, 2897; Leung et al. (2023) Cell 186, 2880). These works shed light on the special protein composition, unique for sperm compared to other cilia (which was studied in Gui et al. (2021) Cell 184, 5791), and correlated the structural information to functional analysis of newly characterized mutants (ciliopathy patients) caused by defect of these proteins. Enormous success of single particle cryo-EM work enabled these structure-function correlation.

This work is heading the same direction, addressing structural characterization of mammalian ciliary doublet structure from sperm and epithelial cells. This work contains several novel findings based on high resolution structures of mammalian ciliary doublet microtubules from various organs and various species. The major results of this work are (1) comparison of ciliary structures with conclusion that sperm cilia are most different with 34 unique proteins, while epithelial cilia are similar to each other, (2) atomic resolution reconstruction of the novel 3D structure of RS3 (the third radial spoke), which was not known, and identification of AKs and PKA, (3) identification of TRiC chaperone at the radial spoke, which was not identified due to lack of resolution in the past, and (4) alternative structure of dynein f, which is likely pre-powerstroke conformation at atomic resolution.

In the viewpoint of this reviewer, (1) provides additional protein composition information, after precise analysis by Zhou et al. and Leung et al. revealing most of the proteins. This additional protein information will be attractive for cilia researchers. (2) and (3) are exciting findings of enzymes, which fit to the functions we expected but could not be located until now, and lead us to nice hypothetical stories how ATP is regenerated and how chaperone works at cilia. (4) present higher resolution view of dynein f conformational change than the past work by Lin and Nicastro (2018) Science 360, 1. Since this work, despite of beautiful illustrations and abundant information, is limited to structural studies of missing pieces with speculative stories solely based on the structure, this reviewer thinks it is suitable for journals specific for molecular structural biology, such as Nature Structural and Molecular Biology (as the authors published in the past NSMB (2022) 29, 483), Molecular Cell or EMBO Journal. With additional studies to prove functions of TRiC, AK or PKA, this work could be attractive for wide readers of Nature. Nevertheless this reviewer would advise the authors for possible revision to improve the manuscript.

We thank the reviewer for their positive overall evaluation of our work. We are keen to explore the possible functions of TRiC in sperm, but these are challenging experiments to do properly. First, there is no reliable, specific inhibitor of TRiC – the compound HSF1A (Neef et al 2014 PMID 25437552) was reported to be one such inhibitor, but we consulted the authors of that study who cautioned against using it. Second, genetically ablating TRiC will likely lead to widespread effects beyond sperm (indeed, the International Mouse Phenotyping Consortium reports that knockout of CCT subunits is embryonic lethal).

In contrast to TRiC, functions of adenylate kinases and PKA in motile cilia have been proven extensively in the literature, particularly for sperm. Genetic disruptions in the genes for AK7 (Fernandez-Gonzalez et al 2009 PMID 18776131, Lores et al 2018 PMID 29365104, Xiang et al 2022 PMID 34854019, Chang et al 2024 PMID 38492416), AK8 (Wu et al 2024 PMID 38761355), and AK9 (Sha et al 2023 PMID 37713809, O'Callaghan et al 2023 PMID 37812723, Wu et al 2024 PMID 38761355) lead to infertility in both humans and mice (and, for AK9, in cattle). Biochemical evidence shows that demembrated sperm can reactivate in the presence of ADP, and that this behavior is lost when AKs are specifically inhibited with the small molecule Ap5A (Schoff

et al 1989 PMID 259368). Consistent with this, a recent study reported decreased ATP levels in AK8 -/- and AK9 -/- sperm (Wu et al 2024 PMID 38761355). We have cited these studies in the Main Text and in Supplementary Table 6 (renumbered from the original Supplementary Table 7).

Likewise, multiple lines of evidence point to PKA as a major mediator of cAMP signaling in sperm, particularly during capacitation – a series of biochemical changes occurring in the female reproductive tract that render sperm capable of fusing with the egg. Two of the hallmarks of capacitation are an increase in protein phosphorylation and activation of motility. Both phenomena are inhibited when sperm are treated with PKA inhibitors (Visconti et al 1995 PMID 7538069, O’Flaherty et al 2004 PMID 14997001, Luconi et al 2005 PMID 15342355, Morgan et al 2008 PMID 19074277, Battistone et al 2013 PMID 23630234). Furthermore, sperm from mice lacking the C α 2 isoform of the PKA catalytic subunit (the only isoform present in mature mouse sperm) are infertile (Nolan et al 2004 PMID 15340140). We have added these citations to the Main Text in the section discussing PKA functions in mammalian sperm.

By showing that AKs and PKA are physically anchored to radial spokes, our study provides a structural mechanism to explain how these enzymes function in the sperm tail and in cilia more broadly.

Major points:

Many newly found proteins are only briefly mentioned. For example, in lines 131 and 132. The authors should be able to describe their conformation, interaction with neighboring proteins and possible functions.

Given space and readability concerns, it’s not possible to describe the conformation and interactions of each newly identified protein in the Main Text. Instead, we have added information about the structure, location, and interaction network of each individual protein to Data S1.

Line132-135, Fig.2b: relationship between the data in this study and ciliopathy is not clear. This reviewer could not find the correlation between sperm specific proteins and sperm specific diseases. More explanation needed.

We apologize for this – we have corrected the color scheme in Fig. 2 to reflect the fact that some sperm-specific proteins (CCDC63, DNAH8, DNAH17, and RSPH6A) show sperm-specific phenotypes.

Note that most of the sperm-specific proteins newly identified in this study (e.g. EFCAB5, STKLD1, TEX47) have not been functionally evaluated either in knockout mice or human patients. Thus, they are colored white in Fig. 2b.

Supplementary Table 2 and Line137-: RS3 molecular modeling

This reviewer is not fully convinced that protein identification can be done at >4Å resolution. In Suppl. Table 2, mostly structure and abundance of protein in cilia are used as the basis. Can the author provide density maps used for chain tracing in EMDB?

To address the comments of all three reviewers, we have generated a supplementary document (Data S1) that contains a separate page for each newly identified protein. Each page includes the database ID of the sequence used for modeling, the location of the protein in the 96-nm repeat, its abundance in the bovine sperm proteome, a model-in-map figure, a correlation coefficient, and the rationale for protein assignment (including identification strategies and supporting evidence from the literature).

As requested, we have deposited locally refined maps to the EMDB. Accession codes are provided in the “Data availability” section and in Supplementary Tables 1 and 2.

Can they compare mammalian cilia and *Chlamydomonas* by proteomics and show absence of RS3 proteins in *Chlamydomonas*, which lacks RS3?

and

Line184-190: Do these molecules have homologues in *Chlamydomonas*? If not, do mammalian cilia need more ATP than protists?

We have expanded Supplementary Table 5 to include putative *Chlamydomonas reinhardtii* orthologs for each newly identified protein. Consistent with the absence of RS3 in *C. reinhardtii*, most RS3 stalk and head subunits lack orthologs in *C. reinhardtii*. The only exceptions are malate dehydrogenase and protein kinase A, which are both common enzymes likely to exist outside the axoneme. *C. reinhardtii* FAP75 (Cre06.g249900) shows some sequence homology to AK7, but their AlphaFold predictions differ and FAP75 likely localizes to the central apparatus (Rao et al 2024 PMID 38140814).

Actively-swimming sea urchin sperm (which have similar RSs to mammals and likely have AKs) do consume more ATP than *Chlamydomonas* axonemes ($\sim 2.4 \times 10^6$ ATP/s for sperm versus $\sim 9.7 \times 10^5$ ATP/s for *Chlamydomonas*), which correlates with the fact that sea urchin sperm flagella have more dyneins than *Chlamydomonas* flagella (Chen et al 2015 PMID 26682814). Given that mammalian sperm are even longer and would have even more dyneins than sea urchin sperm, we predict that their ATP consumption would be even higher.

Line196-200: open and closed states of AKs

How do the authors think two conformations co-exist? Do they show two stages of catalytic cycle? Or just fluctuation? If the authors think it is catalytic, can they shift equilibrium between two states biochemically (by adding phosphate?) and prove it by cryo-EM and classification?

In the case of AK8, the open conformation of domain II appears to be stabilized by contacts with RGS22 and PPIL6. Likewise, open conformations of domains I and IV in AK9 are stabilized by binding of LRRC23. Whether these domains can cycle between open and closed conformations is unclear. The remaining catalytic domains of AK7, AK8, and AK9 are in principle unrestricted and would be able to transition between open and closed conformations during catalysis. We have modified the text to clarify these possibilities, emphasizing that they remain to be tested in the future.

Examining conformational transitions of AKs within the context of the axoneme using cryo-EM will be an interesting but challenging future direction. Our current structures are derived from >40,000 images collected over >10 imaging sessions, and such studies would require collecting similar amounts of data for samples treated with several nucleotide concentrations/combinations. We therefore believe such experiments are beyond the scope of the current work and are best performed as separate follow-up analyses.

Line329-330 and Supplementary Table 8: It is not clear how this table presents earlier or later expression. More explanation is needed.

Supplementary Table 8 uses information from Kaye et al 2024 (PMID: 38287033) to identify genes that are expressed early or late in spermatogenesis. This study performed single-cell RNA sequencing on three distinct cell types, each representing a unique stage of mouse spermatogenesis: spermatogonia (an early cell type), spermatocytes, and round spermatids (a late cell type). The genes were subsequently clustered into six classes, numbered 1-6, based on their relative RNA-seq signal within each cell type. Classes with lower numbers (e.g. 1 and 2) include genes predominantly expressed in spermatogonia (and therefore early in spermatogenesis) whereas classes with higher numbers (e.g. 5 and 6) include genes most highly expressed

in round spermatids, signifying their activation later in spermatogenesis. We have added this information to the legend for Supplementary Table 7 (renumbered from 8 in the original submission).

The proteomics analysis was carried out using different instruments and programs between species/organs to be compared. This raises severe concern. One instrument, Velos Pro, is more than 10 years old instrument, while the other, Eclipse, is the highest end MS and, according to an MS expert, has ~10 times sensitivity compared to Velos Pro. This difference is hard to compensate. They also used two different programs, Mascot and Sequest, for analysis, as well as referring an old work using a different platform. This is not optimum setting to compare protein component between organs and species. This reviewer recommend to analyze MS data using the same platform (ideally Eclipse).

We agree that the Velos Pro is older and less sensitive than the Eclipse. However, as we are not intending the mass spectrometry data to be quantitative across cell/tissue types, the differences in data collection and data analysis do not greatly affect our conclusions. The mass spectrometry data is used only to analyze the composition of the sample used for structural analysis and to support the assignment of a protein as being sperm-specific or more broadly distributed across cilia types. It is never used as the only means to make these assignments, for which we also used tissue expression and single-cell RNA-seq data from the Human Protein Atlas (now added to Supplementary Table 5, renumbered from the original Supplementary Table 6).

Importantly, differences in proteomes across cilia types were not used to assign protein identity in our model of the bovine sperm DMT. We apologize if this was unclear in the original Supplementary Table 5. We have clarified this by removing all statements about relative abundance in the new Data S1 document, which only references the iBAQ value in the bovine sperm mass spectrometry data.

It was also necessary to perform mass spectrometry experiments at different sites and different times since we wanted the proteomes of cilia isolated from tissue (i.e. brain, oviduct, and trachea) to come from the exact samples used for cryo-EM. Repeating the mass spectrometry would, in some instances, require using a different sample from the one used to prepare cryo-EM grids.

Minor points:

Line84: Remarkable consistency with our SPA structure

Zimmermann et al. (2023) EMBO J. 42, e112466 claims ODA structure by cryo-ET needed remodeling from SPA structures. Did the author find no such slight difference between SPA from sprayed axoneme and cryo-ET of intact axoneme?

The paper by Zimmermann and colleagues nicely shows that there can be subtle differences between the ODA conformations observed by SPA and cryo-ET due to the loss of interactions with the neighboring DMT. The motor domains in our SPA structures are not sufficiently well-resolved to do a similar careful analysis, we have therefore rephrased the opening paragraph to instead focus on the consistency of the structures near the microtubule surface.

Line126: new proteins across nearly every axonemal complex, including an ARMH1 subcomplex that is distributed asymmetrically around the axoneme

Did author find ARMH1 for every doublet, while cryo-ET in the past work showed asymmetrical arrangement? What is the reason of the past work missing ARMH1? Did they conduct classification to map the 96nm repeat with and without ARMH1?

In the initial consensus maps, we saw weak density for the ARMH1 subcomplex. 3D classification using a mask derived from previous cryoET map (EMD-27452 from Chen et al 2023 PMID 36593309) allowed us to recover subsets of particles with and without ARMH1. Comparison with doublet-specific subtomogram

averages (from Chen et al 2023 PMID 36593309) suggests that ARMH1 is distributed asymmetrically. We have summarized the processing scheme for ARMH1 in a new Supplementary Figure 7.

Line263: How was pre-powerstroke conformation made? Did the author mix ATP in the specimen? Or was it activated by endogenous ATPs?

The addition of 1 mM ATP to our samples is likely what enriched for the pre-state. The previous cryo-EM reconstruction of the 96-nm repeat from human respiratory cilia (EMD-35888 and PDB 8J07 from Walton & Gui et al 2023 PMID 37258679) was derived from axonemes not treated with ATP, which explains why it captured dyneins in the post-state.

Reviewer Reports on the First Revision:

Referee #1 (Remarks to the Author):

The authors have addressed all my concerns in their revised manuscript. I especially commend them for the well-organized supplementary information that gives additional details on cryo-EM processing and the process for their protein identification. This work is important as it reveals the common core of axonemes from cilia with diverse waveforms (shedding light on the evolutionary process for functional diversification) , it identifies and elucidates the structure and assembly of a large number of proteins unique to the sperm axoneme, while also shedding new light on the organization of the enzymatic machinery that generates ATP, among other interesting findings. The work will be of interest to the cilia field, scientists interested in microtubules and the cytoskeleton, structural biology and scientists interested in large protein assemblies.

Referee #2 (Remarks to the Author):

Overall, the authors have done a satisfactory job addressing the comments. Data S1 showing the data processing and identification strategies is a very nice addition. I have a couple of further comments:

1) It is now clear how the extra densities in the adenylate kinase structure were displayed (Extended Data Figure 8d). However, displaying density within 3-Å of the ligand atoms of the superposed AK crystal structure is potentially problematic, because the reader does not know the threshold level of the map before filtering (i.e. it could be a very low threshold that fills the volume with noise, which is then filtered around the putative ligand atoms). One way to circumvent this would be show the full map with the putative ligand density colored, in addition to (or instead of) the map displaying only density within 3-Å of the ligand atoms from the crystal structure.

2) I find the depiction of the structural changes in dynein-f in Rebuttal Figure 6 much clearer than in Figure 6. Perhaps Rebuttal Figure 6 could be used as a basis for the main figure.

3) I appreciate the authors changing their power stroke nomenclature to reflect the fact that the dyneins are not bound to their track and so can't be in the state before and after they do work. However, the terms 'pre-state' and 'post-state' don't make much sense without further explanation. Possible solutions could be to i) use 'pre-stroke state' and 'post-stroke state' (which still removes the problematic word 'power'), or ii) provide an explanation.

Referee #3 (Remarks to the Author):

The authors addressed mostly requests from the reviewers. Especially their presentation of protein assignment and fitting in Data S1 is quite convincing. Additional description of their cryo-EM data analysis is in detail. The readers will be able to have overview of their analysis.

Their presentation of MS proteomics is still to be improved. Supplementary Tables 3 and 5 is presented without information of different MS instrumentation. Some proteins (such as CAMK4) were detected in bovine cilia, but not in human cilia. Can we exclude the possibility that they were not detected in human cilia by relatively old instrumentation, while a high-end MS instrument detected them in bovine cilia?

This reviewer suggests to add an additional figure of MS instrumentation for each cilia sample, similarly to Supplementary Table 1, before Supple Table 3.

After addressing this point, this reviewer would support publication of this manuscript.

Author Rebuttals to First Revision:**Referee #1**

The authors have addressed all my concerns in their revised manuscript. I especially commend them for the well-organized supplementary information that gives additional details on cryo-EM processing and the process for their protein identification. This work is important as it reveals the common core of axonemes from cilia with diverse waveforms (shedding light on the evolutionary process for functional diversification) , it identifies and elucidates the structure and assembly of a large number of proteins unique to the sperm axoneme, while also shedding new light on the organization of the enzymatic machinery that generates ATP, among other interesting findings. The work will be of interest to the cilia field, scientists interested in microtubules and the cytoskeleton, structural biology and scientists interested in large protein assemblies.

Referee #2

Overall, the authors have done a satisfactory job addressing the comments. Data S1 showing the data processing and identification strategies is a very nice addition. I have a couple of further comments:

1) It is now clear how the extra densities in the adenylate kinase structure were displayed (Extended Data Figure 8d). However, displaying density within 3-Å of the ligand atoms of the superposed AK crystal structure is potentially problematic, because the reader does not know the threshold level of the map before filtering (i.e. it could be a very low threshold that fills the volume with noise, which is then filtered around the putative ligand atoms). One way to circumvent this would be show the full map with the putative ligand density colored, in addition to (or instead of) the map displaying only density within 3-Å of the ligand atoms from the crystal structure.

Thank you for the suggestion. We added panels (Extended Data Figure 8c) showing maps of the AKs with putative ligand density colored.

2) I find the depiction of the structural changes in dynein-f in Rebuttal Figure 6 much clearer than in Figure 6. Perhaps Rebuttal Figure 6 could be used as a basis for the main figure.

We have incorporated the panels from Rebuttal Figure 6 into Figure 5 (previously Figure 6).

3) I appreciate the authors changing their power stroke nomenclature to reflect the fact that the dyneins are not bound to their track and so can't be in the state before and after they do work. However, the terms 'pre-state' and 'post-state' don't make much sense without further explanation. Possible solutions could be to i) use 'pre-stroke state' and 'post-stroke state' (which still removes the problematic word 'power'), or ii) provide an explanation.

We have changed the terms to pre-stroke and post-stroke states.

Referee #3

The authors addressed mostly requests from the reviewers. Especially their presentation of protein assignment and fitting in Data S1 is quite convincing. Additional description of their cryo-EM data analysis is in detail. The readers will be able to have overview of their analysis.

Their presentation of MS proteomics is still to be improved. Supplementary Tables 3 and 5 is presented without information of different MS instrumentation. Some proteins (such as CAMK4) were detected in bovine cilia, but not in human cilia. Can we exclude the possibility that they were not detected in human cilia by relatively old instrumentation, while a high-end MS instrument detected them in bovine cilia? This reviewer suggests to add an additional figure of MS instrumentation for each cilia sample, similarly to Supplementary Table 1, before Supple Table 3.

We have added a new table summarizing the instrumentation used to collect MS data for different mammalian cilia and specifying which datasets were newly collected for this study. This table is now Supplementary Table 3 and subsequent tables have been renumbered.

After addressing this point, this reviewer would support publication of this manuscript.